# KerGM: Kernelized Graph Matching

**Zhen Zhang[1], Yijian Xiang[1], Lingfei Wu[2], Bing Xue[1], Arye Nehorai[1]**
[1]Washington University in St. Louis
[2]IBM Research
[1]{zhen.zhang, yijian.xiang, xuebing, nehorai}@wustl.edu
[2]lwu@email.wm.edu

## Abstract

Graph matching plays a central role in such fields as computer vision, pattern recognition, and bioinformatics. Graph matching problems can be cast as two types of quadratic assignment problems (QAPs): Koopmans-Beckmann's QAP or Lawler's QAP. In our paper, we provide a unifying view for these two problems by introducing new rules for array operations in Hilbert spaces. Consequently, Lawler's QAP can be considered as the Koopmans-Beckmann's alignment between two arrays in reproducing kernel Hilbert spaces (RKHS), making it possible to efficiently solve the problem without computing a huge affinity matrix. Furthermore, we develop the entropy-regularized Frank-Wolfe (EnFW) algorithm for optimizing QAPs, which has the same convergence rate as the original FW algorithm while dramatically reducing the computational burden for each outer iteration. We conduct extensive experiments to evaluate our approach, and show that our algorithm significantly outperforms the state-of-the-art in both matching accuracy and scalability.

## 1 Introduction

Graph matching (GM), which aims at finding the optimal correspondence between nodes of two given graphs, is a longstanding problem due to its nonconvex objective function and binary constraints. It arises in many applications, ranging from recognizing actions [3, 13] to identifying functional orthologs of proteins [11, 41]. Typically, GM problems can be formulated as two kinds of quadratic assignment problems (QAPs): Koopmans-Beckmann's QAP [18] or Lawler's QAP [22]. Koopman-Beckmann's QAP is the structural alignment between two weighted adjacency matrices, which, as a result, can be written as the standard Frobenius inner product between two $n \times n$ matrices, where $n$ denotes the number of nodes. However, Koopmans-Beckmann's QAP cannot incorporate complex edge attribute information, which is usually of great importance in characterizing the relation between nodes. Lawler's QAP can tackle this issue, because it attempts to maximize the overall similarity that well encodes the attribute information. However, the key concern of the Lawler's QAP is that it needs to estimate the $n^2 \times n^2$ pairwise affinity matrix, limiting its application to very small graphs.

In our work, we derive an equivalent formulation of Lawler's QAP, based on a very mild assumption that edge affinities are characterized by kernels [15, 34]. After introducing new rules for array operations in Hilbert spaces, named as $\mathcal{H}-$operations, we rewrite Lawler's QAP as the Koopmann-Beckmann's alignment between two arrays in a reproducing kernel Hilbert space (RKHS), which allows us to solve it without computing the huge affinity matrix. Taking advantage of the $\mathcal{H}-$operations, we develop a path-following strategy for mitigating the local maxima issue of QAPs. In addition to the kernelized graph matching (KerGM) formulation, we propose a numerical optimization algorithm, the entropy-regularized Frank-Wolfe (EnFW) algorithm, for solving large-scale QAPs. The EnFW has the same convergence rate as the original Frank-Wolfe algorithm, with far less computational burden in each iteration. Extensive experimental results show that our KerGM, together with the EnFW algorithm, achieves superior performance in both matching accuracy and scalability.

**Related Work:** In the past forty years, a myriad of graph matching algorithms have been proposed [8], most of which focused on solving QAPs. Previous work [2, 14, 21] approximated the quadratic term with a linear one, which consequently can be solved by standard linear programming solvers. In [36], several convex relaxation methods are proposed and compared. It is known that convex relaxations can achieve global convergence, but usually perform poorly because the final projection step separates the solution from the original QAP. Concave relaxations [29, 28] can avoid this problem since their outputs are just permutation matrices. However, concave programming [4] is NP-hard, which limits its applications. In [45], a seminal work termed the "path-following algorithm" was proposed, which leverages both the above relaxations via iteratively solving a series of optimization problems that gradually changed from convex to concave. In [27, 38, 39, 44], the path following strategy was further extended and improved. However, all the above algorithms, when applied to Lawler's QAP, need to compute the $n^2 \times n^2$ affinity matrix. To tackle the challenge, in [48], the authors elegantly factorized the affinity matrix into the Kronecker product of smaller matrices. However, it still cannot be well applied to large dense graphs, since it scales cubically with the number of edges. Beyond solving the QAP, there are interesting works on doing graph matching from other perspectives, such as probabilistic matching[46], hypergraph matching [24], and multigraph matching [42]. We refer to [43] for a survey of recent advances.

**Organization:** In Section 2, we introduce the background, including Koopmans-Beckmann's and Lawler's QAPs, and kernel functions and its reproducing kernel Hilbert space. In Section 3, we present the proposed rules for array operations in Hilbert space. Section 4 and Section 5 form the core of our work, where we develop the kernelized graph matching, together with the entropy-regularized Frank-Wolfe optimizaton algorithm. In Section 6, we report the experimental results. In the supplementary material, we provide proofs of all mathematical results in the paper, along with further technical discussions and more experimental results.

## 2 Background

### 2.1 Quadratic Assignment Problems for Graph Matching

Let $\mathcal{G} = \{\boldsymbol{A}, \mathcal{V}, \boldsymbol{P}, \mathcal{E}, \boldsymbol{Q}\}$ be an undirected, attributed graph of $n$ nodes and $m$ edges, where $\boldsymbol{A} \in \mathbb{R}^{n \times n}$ is the adjacency matrix, $\mathcal{V} = \{v_i\}_{i=1}^n$ and $\boldsymbol{P} = [\vec{\boldsymbol{p}}_1, \vec{\boldsymbol{p}}_2, ..., \vec{\boldsymbol{p}}_n] \in \mathbb{R}^{d_N \times n}$ are the respective node set and node attributes matrix, and $\mathcal{E} = \{e_{ij} | v_i \text{ and } v_j \text{ are connected}\}$ and $\boldsymbol{Q} = [\vec{\boldsymbol{q}}_{ij} | e_{ij} \in \mathcal{E}] \in \mathbb{R}^{d_E \times m}$ are the respective edge set and edge attributes matrix. Given two graphs $\mathcal{G}_1 = \{\boldsymbol{A}_1, \mathcal{V}_1, \boldsymbol{P}_1, \mathcal{E}_1, \boldsymbol{Q}_1\}$ and $\mathcal{G}_2 = \{\boldsymbol{A}_2, \mathcal{V}_2, \boldsymbol{P}_2, \mathcal{E}_2, \boldsymbol{Q}_2\}$ of $n$ nodes[1], the GM problem aims to find a correspondence between nodes in $\mathcal{V}_1$ and $\mathcal{V}_2$ which is optimal in some sense.

For **Koopmans-Beckmann's QAP** [18], the optimality refers to the Frobenius inner product maximization between two adjacency matrices after permutation, i.e.,

$$\max \langle \boldsymbol{A}_1 \boldsymbol{X}, \boldsymbol{X} \boldsymbol{A}_2 \rangle_{\mathrm{F}} \quad \text{s.t. } \boldsymbol{X} \in \mathcal{P} = \{\boldsymbol{X} \in \{0,1\}^{n \times n} | \boldsymbol{X} \vec{\boldsymbol{1}} = \vec{\boldsymbol{1}}, \boldsymbol{X}^T \vec{\boldsymbol{1}} = \vec{\boldsymbol{1}}\}, \quad (1)$$

where $\langle \boldsymbol{A}, \boldsymbol{B} \rangle_{\mathrm{F}} = \mathrm{tr}(\boldsymbol{A}^T \boldsymbol{B})$ is the Frobenius inner product. The issue with (1) is that it ignores the complex edge attributes, which are usually of particular importance in characterizing graphs.

For **Lawler's QAP** [22], the optimality refers to the similarity maximization between the node attribute sets and between the edge attribute sets, i.e.,

$$\max \sum_{v_i^1 \in \mathcal{V}_1, v_a^2 \in \mathcal{V}_2} k^N(\vec{\boldsymbol{p}}_i^1, \vec{\boldsymbol{p}}_a^2) \boldsymbol{X}_{ia} + \sum_{e_{ij}^1 \in \mathcal{E}_1, e_{ab}^2 \in \mathcal{E}_2} k^E(\vec{\boldsymbol{q}}_{ij}^1, \vec{\boldsymbol{q}}_{ab}^2) \boldsymbol{X}_{ia} \boldsymbol{X}_{jb} \quad \text{s.t. } \boldsymbol{X} \in \mathcal{P}, \quad (2)$$

where $k^N$ and $k^E$ are the node and edge similarity measurements, respectively. Furthermore, (2) can be rewritten in compact form:

$$\max \langle \boldsymbol{K}^N, \boldsymbol{X} \rangle_{\mathrm{F}} + \mathrm{vec}(\boldsymbol{X})^T \boldsymbol{K} \mathrm{vec}(\boldsymbol{X}) \quad \text{s.t. } \boldsymbol{X} \in \mathcal{P}, \quad (3)$$

where $\boldsymbol{K}^N \in \mathbb{R}^{n \times n}$ is the node affinity matrix, $\boldsymbol{K}$ is an $n^2 \times n^2$ matrix, defined such that $\boldsymbol{K}_{ia,jb} = k^E(\vec{\boldsymbol{q}}_{ij}^1, \vec{\boldsymbol{q}}_{ab}^2)$ if $i \neq j$, $a \neq b$, $e_{ij}^1 \in \mathcal{E}_1$, and $e_{ab}^2 \in \mathcal{E}_2$, otherwise, $\boldsymbol{K}_{ia,jb} = 0$. It is well known that Koopmans-Beckmann's QAP is a special case of Lawler's QAP if we set $\boldsymbol{K} = \boldsymbol{A}_2 \otimes \boldsymbol{A}_1$ and $\boldsymbol{K}^N = \boldsymbol{0}_{n \times n}$. The issue of (3) is that the size of $\boldsymbol{K}$ scales quadruply with respect to $n$, which precludes its applications to large graphs. In our work, we will show that Lawler's QAP can be written in the Koopmans-Beckmann's form, which can avoid computing $\boldsymbol{K}$.

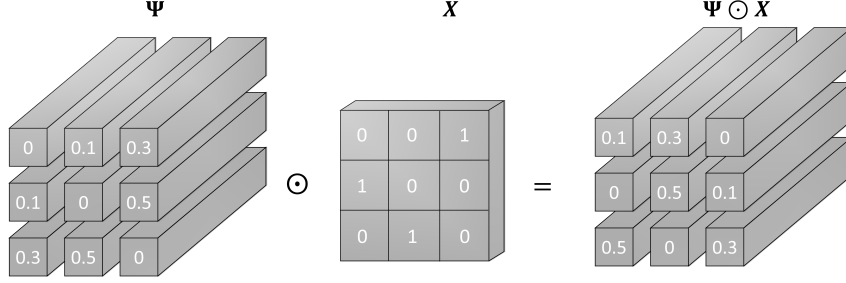

Figure 1: Visualization of the operation $\mathbf{\Psi} \odot \mathbf{X}$.

## 2.2   Kernels and reproducing kernel Hilbert spaces

Given any set $\mathcal{X}$, a kernel $k : \mathcal{X} \times \mathcal{X} \to \mathbb{R}$ is a function for quantitatively measuring the affinity between objects in $\mathcal{X}$. It satisfies that there exist a Hilbert space, $\mathcal{H}$, and an (implicit) feature map $\psi : \mathcal{X} \to \mathcal{H}$, such that $k(q^1, q^2) = \langle \psi(q^1), \psi(q^2) \rangle_{\mathcal{H}}, \forall q^1, q^2 \in \mathcal{X}$. The space $\mathcal{H}$ is the reproducing kernel Hilbert space associated with $k$.

Note that if $\mathcal{X}$ is a Euclidean space i.e., $\mathcal{X} = \mathbb{R}^d$, many similarity measurement functions are kernels, such as $\exp(-\|\vec{q}^1 - \vec{q}^2\|_2^2)$, $\exp(-\|\vec{q}^1 - \vec{q}^2\|_2)$, and $\langle \vec{q}^1, \vec{q}^2 \rangle, \forall \vec{q}^1, \vec{q}^2 \in \mathbb{R}^d$.

## 3   $\mathcal{H}$-operations for arrays in Hilbert spaces

Let $\mathcal{H}$ be any Hilbert space, coupled with the inner product $\langle \cdot, \cdot \rangle_{\mathcal{H}}$ taking values in $\mathbb{R}$. Let $\mathcal{H}^{n \times n}$ be the set of all $n \times n$ arrays in $\mathcal{H}$, and let $\mathbf{\Psi}, \mathbf{\Xi} \in \mathcal{H}^{n \times n}$, i.e., $\mathbf{\Psi}_{ij}, \mathbf{\Xi}_{ij} \in \mathcal{H}, \forall i, j = 1, 2, ..., n$. Analogous to matrix operations in Euclidean spaces, we make the following addition, transpose, and multiplication rules ($\mathcal{H}$-operations), i.e.,$\forall \mathbf{X} \in \mathbb{R}^{n \times n}$, and we have

1.   $\mathbf{\Psi} + \mathbf{\Xi}, \mathbf{\Psi}^T \in \mathcal{H}^{n \times n}$, where $[\mathbf{\Psi} + \mathbf{\Xi}]_{ij} \triangleq \mathbf{\Psi}_{ij} + \mathbf{\Xi}_{ij} \in \mathcal{H}$ and $[\mathbf{\Psi}^T]_{ij} \triangleq \mathbf{\Psi}_{ji} \in \mathcal{H}$.

2.   $\mathbf{\Psi} * \mathbf{\Xi} \in \mathbb{R}^{n \times n}$, where $[\mathbf{\Psi} * \mathbf{\Xi}]_{ij} \triangleq \sum_{k=1}^{n} \langle \mathbf{\Psi}_{ik}, \mathbf{\Xi}_{kj} \rangle_{\mathcal{H}} \in \mathbb{R}$.

3.   $\mathbf{\Psi} \odot \mathbf{X}, \mathbf{X} \odot \mathbf{\Psi} \in \mathcal{H}^{n \times n}$, where $[\mathbf{\Psi} \odot \mathbf{X}]_{ij} \triangleq \sum_{k=1}^{n} \mathbf{\Psi}_{ik} \mathbf{X}_{kj} = \sum_{k=1}^{n} \mathbf{X}_{kj} \mathbf{\Psi}_{ik} \in \mathcal{H}$ and $[\mathbf{X} \odot \mathbf{\Psi}]_{ij} \triangleq \sum_{k=1}^{n} \mathbf{X}_{ik} \mathbf{\Psi}_{kj} \in \mathcal{H}$.

Note that if $\mathcal{H} = \mathbb{R}$, all the above degenerate to the common operations for matrices in Euclidean spaces. In Fig. 1, we visualize the operation $\mathbf{\Psi} \odot \mathbf{X}$, where we let $\mathcal{H} = \mathbb{R}^d$, let $\mathbf{\Psi}$ be a $3 \times 3$ array in $\mathbb{R}^d$, and let $\mathbf{X}$ be a $3 \times 3$ permutation matrix. It is easy to see that $\mathbf{\Psi} \odot \mathbf{X}$ is just $\mathbf{\Psi}$ after column-permutation.

As presented in the following corollary, the multiplication $\odot$ satisfy the combination law.

**Corollary 1.** $\forall \mathbf{X}, \mathbf{Y} \in \mathbb{R}^{n \times n}$, $\mathbf{\Psi} \odot \mathbf{X} \odot \mathbf{Y} = \mathbf{\Psi} \odot (\mathbf{X}\mathbf{Y})$, and $\mathbf{Y} \odot (\mathbf{X} \odot \mathbf{\Psi}) = (\mathbf{Y}\mathbf{X}) \odot \mathbf{\Psi}$.

Based on the $\mathcal{H}$-operations, we can construct the Frobenius inner product on $\mathcal{H}^{n \times n}$.

**Proposition 1.** *Define the function* $\langle \cdot, \cdot \rangle_{\mathrm{F}_{\mathcal{H}}} : \mathcal{H}^{n \times n} \times \mathcal{H}^{n \times n} \to \mathbb{R}$ *such that* $\langle \mathbf{\Psi}, \mathbf{\Xi} \rangle_{\mathrm{F}_{\mathcal{H}}} \triangleq \mathrm{tr}(\mathbf{\Psi}^T * \mathbf{\Xi}) = \sum_{i,j=1}^{n} \langle \mathbf{\Psi}_{ij}, \mathbf{\Xi}_{ij} \rangle_{\mathcal{H}}, \forall \mathbf{\Psi}, \mathbf{\Xi} \in \mathcal{H}^{n \times n}$. *Then* $\langle \cdot, \cdot \rangle_{\mathrm{F}_{\mathcal{H}}}$ *is an inner product on* $\mathcal{H}^{n \times n}$.

As an immediate result, the function $\| \cdot \|_{\mathrm{F}_{\mathcal{H}}} : \mathcal{H}^{n \times n} \to \mathbb{R}$, defined such that $\|\mathbf{\Psi}\|_{\mathrm{F}_{\mathcal{H}}} = \sqrt{\langle \mathbf{\Psi}, \mathbf{\Psi} \rangle_{\mathrm{F}_{\mathcal{H}}}}$, is the Frobenius norm on $\mathcal{H}^{n \times n}$. Next, we introduce two properties of $\langle \cdot, \cdot \rangle_{\mathrm{F}_{\mathcal{H}}}$, which play important roles for developing the convex-concave relaxation of the Lawler's graph matching problem.

**Corollary 2.** $\langle \mathbf{\Psi} \odot \mathbf{X}, \mathbf{\Xi} \rangle_{\mathrm{F}_{\mathcal{H}}} = \langle \mathbf{\Psi}, \mathbf{\Xi} \odot \mathbf{X}^T \rangle_{\mathrm{F}_{\mathcal{H}}}$ *and* $\langle \mathbf{X} \odot \mathbf{\Psi}, \mathbf{\Xi} \rangle_{\mathrm{F}_{\mathcal{H}}} = \langle \mathbf{\Psi}, \mathbf{X}^T \odot \mathbf{\Xi} \rangle_{\mathrm{F}_{\mathcal{H}}}$.

## 4   Kernelized graph matching

Before deriving kernelized graph matching, we first present an assumption.

**Assumption 1.** *We assume that the edge affinity function* $k^E : \mathbb{R}^{d_E} \times \mathbb{R}^{d_E} \to \mathbb{R}$ *is a kernel. That is, there exist both an RKHS,* $\mathcal{H}$, *and an (implicit) feature map,* $\psi : \mathbb{R}^{d_E} \to \mathcal{H}$, *such that* $k^E(\vec{q}^1, \vec{q}^2) = \langle \psi(\vec{q}^1), \psi(\vec{q}^2) \rangle_{\mathcal{H}}, \forall \vec{q}^1, \vec{q}^2 \in \mathbb{R}^{d_E}$.

Note that Assumption 1 is rather mild, since kernel functions are powerful and popular in quantifying the similarity between attributes [47], [19].

For any graph $\mathcal{G} = \{\boldsymbol{A}, \mathcal{V}, \boldsymbol{P}, \mathcal{E}, \boldsymbol{Q}\}$, we can construct an array, $\boldsymbol{\Psi} \in \mathcal{H}^{n \times n}$:

$$\boldsymbol{\Psi}_{ij} = \begin{cases} \psi(\vec{q}_{ij}) \in \mathcal{H}, & \text{if } (v_i, v_j) \in \mathcal{E} \\ 0_{\mathcal{H}} \quad \in \mathcal{H}, & \text{otherwise} \end{cases}, \text{where } 0_{\mathcal{H}} \text{ is the zero vector in } \mathcal{H}. \tag{4}$$

Given two graphs $\mathcal{G}_1$ and $\mathcal{G}_2$, let $\boldsymbol{\Psi}^{(1)}$ and $\boldsymbol{\Psi}^{(2)}$ be the corresponding Hilbert arrays of $\mathcal{G}_1$ and $\mathcal{G}_2$, respectively. Then the edge similarity term in Lawler's QAP (see (2)) can be written as

$$\sum_{e^1_{ij} \in \mathcal{E}_1, e^2_{ab} \in \mathcal{E}_2} k^E(\vec{q}^1_{ij}, \vec{q}^2_{ab}) X_{ia} X_{jb} = \sum_{i,b=1}^n \langle \sum_{j=1}^n \boldsymbol{\Psi}^{(1)}_{ij} X_{jb}, \sum_{a=1}^n X_{ia} \boldsymbol{\Psi}^{(2)}_{ab} \rangle_{\mathcal{H}_{\mathcal{K}}} = \langle \boldsymbol{\Psi}^{(1)} \odot \boldsymbol{X}, \boldsymbol{X} \odot \boldsymbol{\Psi}^{(2)} \rangle_{\mathrm{F}_{\mathcal{H}}},$$

which shares a similar form with (1), and can be considered as the Koopmans-Beckmann's alignment between the Hilbert arrays $\boldsymbol{\Psi}^{(1)}$ and $\boldsymbol{\Psi}^{(2)}$. The last term in (4) is just the Frobenius inner product between two Hilbert arrays after permutation. Adding the node affinity term, we write Laweler's QAP as[2]:

$$\min \ J_{\mathrm{gm}}(\boldsymbol{X}) = -\langle \boldsymbol{K}^N, \boldsymbol{X} \rangle_{\mathrm{F}} - \langle \boldsymbol{\Psi}^{(1)} \odot \boldsymbol{X}, \boldsymbol{X} \odot \boldsymbol{\Psi}^{(2)} \rangle_{\mathrm{F}_{\mathcal{H}}} \quad \text{s.t. } \boldsymbol{X} \in \mathcal{P}. \tag{5}$$

### 4.1 Convex and concave relaxations

The form (5) inspires an intuitive way to develop convex and concave relaxations. To do this, we first introduce an auxiliary function $J_{\mathrm{aux}}(\boldsymbol{X}) = \frac{1}{2}\langle \boldsymbol{\Psi}^{(1)} \odot \boldsymbol{X}, \boldsymbol{\Psi}^{(1)} \odot \boldsymbol{X} \rangle_{\mathrm{F}_{\mathcal{H}}} + \frac{1}{2}\langle \boldsymbol{X} \odot \boldsymbol{\Psi}^{(2)}, \boldsymbol{X} \odot \boldsymbol{\Psi}^{(2)} \rangle_{\mathrm{F}_{\mathcal{H}}}$. Applying Corollary 1 and 2, for any $\boldsymbol{X} \in \mathcal{P}$, which satisfies $\boldsymbol{X}\boldsymbol{X}^T = \boldsymbol{X}^T\boldsymbol{X} = \boldsymbol{I}$, we have

$$J_{\mathrm{aux}}(\boldsymbol{X}) = \frac{1}{2}\langle \boldsymbol{\Psi}^{(1)}, \boldsymbol{\Psi}^{(1)} \odot (\boldsymbol{X}\boldsymbol{X}^T) \rangle_{\mathrm{F}_{\mathcal{H}}} + \frac{1}{2}\langle \boldsymbol{\Psi}^{(2)}, (\boldsymbol{X}^T\boldsymbol{X}) \odot \boldsymbol{\Psi}^{(2)} \rangle_{\mathrm{F}_{\mathcal{H}}} = \frac{1}{2}\|\boldsymbol{\Psi}^{(1)}\|_{\mathrm{F}_{\mathcal{H}}}^2 + \frac{1}{2}\|\boldsymbol{\Psi}^{(2)}\|_{\mathrm{F}_{\mathcal{H}}}^2,$$

which is always a constant. Introducing $J_{\mathrm{aux}}(\boldsymbol{X})$ to (5), we obtain convex and concave relaxations:

$$J_{\mathrm{vex}}(\boldsymbol{X}) = J_{\mathrm{gm}}(\boldsymbol{X}) + J_{\mathrm{aux}}(\boldsymbol{X}) = -\langle \boldsymbol{K}^N, \boldsymbol{X} \rangle_{\mathrm{F}} + \frac{1}{2}\|\boldsymbol{\Psi}^{(1)} \odot \boldsymbol{X} - \boldsymbol{X} \odot \boldsymbol{\Psi}^{(2)}\|_{\mathrm{F}_{\mathcal{H}}}^2, \tag{6}$$

$$J_{\mathrm{cav}}(\boldsymbol{X}) = J_{\mathrm{gm}}(\boldsymbol{X}) - J_{\mathrm{aux}}(\boldsymbol{X}) = -\langle \boldsymbol{K}^N, \boldsymbol{X} \rangle_{\mathrm{F}} - \frac{1}{2}\|\boldsymbol{\Psi}^{(1)} \odot \boldsymbol{X} + \boldsymbol{X} \odot \boldsymbol{\Psi}^{(2)}\|_{\mathrm{F}_{\mathcal{H}}}^2. \tag{7}$$

The convexity of $J_{\mathrm{vex}}(\boldsymbol{X})$ is easy to conclude, because the composite function of the squared norm, $\|\cdot\|_{\mathrm{F}_{\mathcal{H}}}^2$, and the linear transformation, $\boldsymbol{\Psi}^{(1)} \odot \boldsymbol{X} - \boldsymbol{X} \odot \boldsymbol{\Psi}^{(2)}$, is convex. We have similarity interpretation for the concavity of $J_{\mathrm{cav}}(\boldsymbol{X})$.

It is interesting to see that the term $\frac{1}{2}\|\boldsymbol{\Psi}^{(1)} \odot \boldsymbol{X} - \boldsymbol{X} \odot \boldsymbol{\Psi}^{(2)}\|_{\mathrm{F}_{\mathcal{H}}}$ in (6) is just the distance between Hilbert arrays. If we set the map $\psi(x) = x$, then the convex relaxation of (1) is recovered (see [1]).

**Path following strategy:** Leveraging these two relaxations [45], we minimize $J_{\mathrm{gm}}$ by successively optimizing a series of sub-problems parameterized by $\alpha \in [0, 1]$:

$$\min J_\alpha(\boldsymbol{X}) = (1-\alpha)J_{\mathrm{vex}}(\boldsymbol{X}) + \alpha J_{\mathrm{cav}}(\boldsymbol{X}) \quad \text{s.t. } \boldsymbol{X} \in \mathcal{D} = \{\boldsymbol{X} \in \mathbb{R}_+^{n \times n} | \boldsymbol{X}\mathbf{1} = \mathbf{1}, \boldsymbol{X}^T\mathbf{1} = \mathbf{1}\}, \tag{8}$$

where $\mathcal{D}$ is the double stochastic relaxation of the permutation matrix set, $\mathcal{P}$. We start at $\alpha = 0$ and find the unique minimum. Then we gradually increase $\alpha$ until $\alpha = 1$. That is, we optimize $J_{\alpha + \triangle \alpha}$ with the local minimizer of $J_\alpha$ as the initial point. Finally, we output the local minimizer of $J_{\alpha=1}$. We refer to [45], [48], and [39] for detailed descriptions and improvements.

**Gradients computation:** If we use the first-order optimization methods, we need only the gradients:

$$\nabla J_\alpha(\boldsymbol{X}) = (1 - 2\alpha)\big[(\boldsymbol{\Psi}^{(1)} * \boldsymbol{\Psi}^{(1)})\boldsymbol{X} + \boldsymbol{X}(\boldsymbol{\Psi}^{(2)} * \boldsymbol{\Psi}^{(2)})\big] - 2(\boldsymbol{\Psi}^{(1)} \odot \boldsymbol{X}) * \boldsymbol{\Psi}^{(2)} - \boldsymbol{K}^N, \tag{9}$$

where $\forall i, j = 1, 2, ..., n$, $[\boldsymbol{\Psi}^{(1)} * \boldsymbol{\Psi}^{(1)}]_{ij} = \sum_{e^1_{ik}, e^1_{kj} \in \mathcal{E}_1} k^E(\vec{q}^1_{ik}, \vec{q}^1_{kj})$; $\forall a, b = 1, 2, ..., n$, $[\boldsymbol{\Psi}^{(2)} * \boldsymbol{\Psi}^{(2)}]_{ab} = \sum_{e^2_{ac}, e^2_{cb} \in \mathcal{E}_2} k^E(\vec{q}^2_{ac}, \vec{q}^2_{cb})$; and $\forall i, a = 1, 2, ..., n$, $[(\boldsymbol{\Psi}^{(1)} \odot \boldsymbol{X}) * \boldsymbol{\Psi}^{(2)}]_{ia} = \sum_{e^1_{ik} \in \mathcal{E}_1, e^2_{ca} \in \mathcal{E}_2} X_{kc} k^E(\vec{q}^1_{ik}, \vec{q}^2_{ca})$. In the supplementary material, we provide compact matrix multiplication forms for computing (9).

## 4.2 Approximate explicit feature maps

Based on the above discussion, we significantly reduce the space cost of Lawler's QAP by avoiding computing the affinity matrix $\boldsymbol{K} \in \mathbb{R}^{n^2 \times n^2}$. However, the time cost of computing gradient with (9) is $O(n^4)$, which can be further reduced by employing the approximate explicit feature maps [33, 40].

For the kernel $k^E : \mathbb{R}^{d_E} \times \mathbb{R}^{d_E} \to \mathbb{R}$, we may find an explicit feature map $\hat{\psi} : \mathbb{R}^{d_E} \to \mathbb{R}^D$, such that

$$\forall \vec{q}^1, \vec{q}^2 \in \mathbb{R}^{d_E}, \ \langle \hat{\psi}(\vec{q}^1), \hat{\psi}(\vec{q}^2) \rangle = \hat{k}^E(\vec{q}^1, \vec{q}^2) \approx k^E(\vec{q}^1, \vec{q}^2). \tag{10}$$

For example, if $k^E(\vec{q}^1, \vec{q}^2) = \exp(-\gamma \|\vec{q}^1 - \vec{q}^2\|_2^2)$, then $\hat{\psi}$ is the Fourier random feature map [33]:

$$\hat{\psi}(\vec{q}) = \sqrt{\frac{2}{D}} \left[ \cos(\omega_1^T \vec{q} + b_1), ..., \cos(\omega_D^T \vec{q} + b_D) \right]^T, \text{ where } \omega_i \sim N(\vec{0}, \gamma^2 \boldsymbol{I}) \text{ and } b_i \sim U[0, 1]. \tag{11}$$

Note that in practice, the performance of $\hat{\psi}$ is good enough for relatively small values of $D$ [47]. By the virtue of explicit feature maps, we obtain a new graph representation $\hat{\boldsymbol{\Psi}} \in (\mathbb{R}^D)^{n \times n}$:

$$\hat{\boldsymbol{\Psi}}_{ij} = \begin{cases} \hat{\psi}(\vec{q}_{ij}) \in \mathbb{R}^D, & \text{if } (v_i, v_j) \in \mathcal{E} \\ \vec{0} \qquad \in \mathbb{R}^D, & \text{otherwise} \end{cases}, \text{ where } \vec{0} \text{ is the zero vector in } \mathbb{R}^D. \tag{12}$$

Its space cost is $O(Dn^2)$. Now computing the gradient-related terms $\hat{\boldsymbol{\Psi}}^\triangle * \hat{\boldsymbol{\Psi}}^\triangle$, $\triangle = (1), (2)$, and $(\hat{\boldsymbol{\Psi}}^{(1)} \odot \boldsymbol{X}) * \hat{\boldsymbol{\Psi}}^{(2)}$ in (9) becomes rather simple. We first slice $\hat{\boldsymbol{\Psi}}^\triangle$ into $D$ matrices $\hat{\boldsymbol{\Psi}}^\triangle(:,:,i) \in \mathbb{R}^{n \times n}$, $i = 1, 2, ..., D$. Then it can be easily shown that

$$\hat{\boldsymbol{\Psi}}^\triangle * \hat{\boldsymbol{\Psi}}^\triangle = \sum_{i=1}^D \hat{\boldsymbol{\Psi}}^\triangle(:,:,i) \hat{\boldsymbol{\Psi}}^\triangle(:,:,i), \text{ and } (\hat{\boldsymbol{\Psi}}^{(1)} \odot \boldsymbol{X}) * \hat{\boldsymbol{\Psi}}^{(2)} = \sum_{i=1}^D \hat{\boldsymbol{\Psi}}^{(1)}(:,:,i) \boldsymbol{X} \hat{\boldsymbol{\Psi}}^{(2)}(:,:,i), \tag{13}$$

whose the first and second term respectively involves $D$ and $2D$ matrix multiplications of the size $n \times n$. Hence, the time complexity is reduced to $O(Dn^3)$. Moreover, gradient computations with (13) are highly parallelizable, which also contributes to scalability.

## 5 Entropy-regularized Frank-Wolfe optimization algorithm

The state-of-the-art method for optimizing problem (8) is the Frank-Wolfe algorithm [29, 25, 37, 49], whose every iteration involves linear programming to obtain optimal direction $\boldsymbol{Y}^*$, i.e.,

$$\boldsymbol{Y}^* = \operatorname{argmin}_{\boldsymbol{Y} \in \mathcal{D}} \langle \nabla J_\alpha(\boldsymbol{X}), \boldsymbol{Y} \rangle_{\mathrm{F}}, \tag{14}$$

which is usually solved by the Hungarian algorithm [20]. Optimizing $J_\alpha$ may need to call the Hungarian algorithm many times, which is quite time-consuming for large graphs. In this work, instead of minimizing $J_\alpha(\boldsymbol{X})$ in (8), we consider the following problem,

$$\min_{\boldsymbol{X}} \quad F_\alpha(\boldsymbol{X}) = J_\alpha(\boldsymbol{X}) + \lambda H(\boldsymbol{X}) \quad \text{s.t. } \boldsymbol{X} \in \mathcal{D}_n, \tag{15}$$

where $\mathcal{D}_n = \{ \boldsymbol{X} \in \mathbb{R}_+^{n \times n} | \boldsymbol{X} \boldsymbol{1} = \frac{1}{n} \boldsymbol{1}, \boldsymbol{X}^T \boldsymbol{1} = \frac{1}{n} \boldsymbol{1} \}$, $H(\boldsymbol{X}) = \sum_{i,j=1}^n \boldsymbol{X}_{ij} \log \boldsymbol{X}_{ij}$ is the negative entropy, and the node affinity matrix $\boldsymbol{K}^N$ in $J_\alpha(\boldsymbol{X})$ (see (5) and (8)) is normalized as $\boldsymbol{K}^N \to \frac{1}{n} \boldsymbol{K}^N$ to balance the node and edge affinity terms. The observation is that if $\lambda$ is set to be small enough, the solution of (15), after being multiplied by $n$, will approximate that of the original QAP (8) as much as possible. We design the entropy-regularized Frank-Wolfe algorithm ("EnFW" for short) for optimizing (15), in each outer iteration of which we solve the following nonlinear problem.

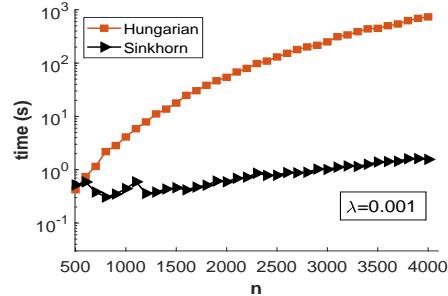

Figure 2: Hungarian vs Sinkhorn.

$$\min \langle \nabla J_\alpha(\boldsymbol{X}), \boldsymbol{Y} \rangle_{\mathrm{F}} + \lambda H(\boldsymbol{Y}) \quad \text{s.t. } \boldsymbol{Y} \in \mathcal{D}_n. \tag{16}$$

Note that (16) can be extremely efficiently solved by the Sinkhorn-Knopp algorithm [10]. Theoretically, the Sinkhorn-Knopp algorithm converges at the linear rate, i.e., $0 < \limsup \|\boldsymbol{Y}_{k+1} - \boldsymbol{Y}^*\|/\|\boldsymbol{Y}_k - \boldsymbol{Y}^*\| < 1$. An empirical comparison between the runtimes of these two algorithms is shown in Fig. 2, where we can see that the Sinkhorn-Knopp algorithm for solving (16) is much faster than the Hungarian algorithm for solving (14).

**The EnFW algorithm description:** We first give necessary definitions. Write the quadratic function $J_\alpha(\boldsymbol{X}+s(\boldsymbol{Y}-\boldsymbol{X})) = J_\alpha(\boldsymbol{X}) + s\langle \nabla J_\alpha(\boldsymbol{X}), \boldsymbol{Y} - \boldsymbol{X}\rangle_{\mathrm{F}} + \frac{1}{2}\mathrm{vec}(\boldsymbol{Y}-\boldsymbol{X})^T \nabla^2 J_\alpha(\boldsymbol{X})\mathrm{vec}(\boldsymbol{Y}-\boldsymbol{X})s^2$. Then, we define the coefficient of the quadratic term as

$$Q(\boldsymbol{X},\boldsymbol{Y}) \triangleq \frac{1}{2}\mathrm{vec}(\boldsymbol{Y}-\boldsymbol{X})^T \nabla^2 J_\alpha(\boldsymbol{X})\mathrm{vec}(\boldsymbol{Y}-\boldsymbol{X}) = \frac{1}{2}\langle \nabla J_\alpha(\boldsymbol{Y}-\boldsymbol{X}), \boldsymbol{Y}-\boldsymbol{X}\rangle_{\mathrm{F}}, \qquad (17)$$

where the second equality holds because $J_\alpha$ is a quadratic function. Next, similar to the original FW algorithm, we define the nonnegative gap function $g(\boldsymbol{X})$ as

$$g(\boldsymbol{X}) \triangleq \langle \nabla J_\alpha(\boldsymbol{X}), \boldsymbol{X}\rangle_{\mathrm{F}} + \lambda H(\boldsymbol{X}) - \min_{\boldsymbol{Y}\in\mathcal{D}_n} \langle \nabla J_\alpha(\boldsymbol{X}), \boldsymbol{Y}\rangle_{\mathrm{F}} + \lambda H(\boldsymbol{Y}). \qquad (18)$$

**Proposition 2.** *If $\boldsymbol{X}^*$ is an optimal solution of* (15)*, then $g(\boldsymbol{X}^*)=0$.*

Therefore, the gap function characterize the necessary condition for optimal solutions. Note that for any $\boldsymbol{X} \in \mathcal{D}_n$, if $g(\boldsymbol{X})=0$, then we say "$\boldsymbol{X}$ is a first-order stationary point". Now with the definitions of $Q(\boldsymbol{X},\boldsymbol{Y})$ and $g(\boldsymbol{X})$, we detail the EnFW procedure in Algorithm 1.

---

**Algorithm 1** The EnFW optimization algorithm for minimizing $F_\alpha$ (15)

---

1: Initialize $\boldsymbol{X}_0 \in \mathcal{D}_n$
2: **while** not converge **do**
3:     Compute the gradient $\nabla J_\alpha(\boldsymbol{X}_t)$ based on (9) or (13),
4:     Obtain the optimal direction $\boldsymbol{Y}_t$ by solving (16), i.e., $\boldsymbol{Y}_t = \mathrm{argmin}_{\boldsymbol{Y}\in\mathcal{D}_n} \langle \nabla J_\alpha(\boldsymbol{X}_t), \boldsymbol{Y}\rangle_{\mathrm{F}} + \lambda H(\boldsymbol{Y})$,
5:     Compute $G_t = g(\boldsymbol{X}_t)$ and $Q_t = Q(\boldsymbol{X}_t, \boldsymbol{Y}_t)$,
6:     Determine the stepsize $s_t$: If $Q_t \le 0$; $s_t = 1$, else $s_t = \min\{G_t/(2Q_t), 1\}$,
7:     Update $\boldsymbol{X}_{t+1} = \boldsymbol{X}_t + s_t(\boldsymbol{Y}_t - \boldsymbol{X}_t)$.
8: **end**
9: Output the solution $\boldsymbol{X}_\alpha^*$.

---

After obtaining the optimal solution path $\boldsymbol{X}_\alpha^*$, $\alpha = 0 : \triangle\alpha : 1$, we discretize $n\boldsymbol{X}_1^*$ by the Hungarian [20] or the greedy discretization algorithm [5] to get the binary matching matrix. We next highlight the differences between the EnFW algorithm and the original FW algorithm: (i) We find the optimal direction by solving a nonlinear convex problem (16) with the efficient Sinkhorn-Knopp algorithm, instead of solving the linear problem (14). (ii) We give an explicit formula for computing the stepsize $s$, instead of making a linear search on $[0,1]$ for optimizing $F_\alpha(\boldsymbol{X}+s(\boldsymbol{Y}-\boldsymbol{X}))$ or estimating the Lipschitz constant of $\nabla F_\alpha$ [32].

### 5.1 Convergence analysis

In this part, we present the convergence properties of the proposed EnFW algorithm, including the sequentially decreasing property of the objective function and the convergence rates.

**Theorem 1.** *The generated objective function value sequence, $\{F_\alpha(\boldsymbol{X}_t)\}_{t=0}$, will decreasingly converge. The generated points sequence, $\{\boldsymbol{X}_t\}_{t=0} \subseteq \mathcal{D}_n \subseteq \mathbb{R}^{n\times n}$, will weakly converge to the first-order stationary point, at the rate $O(\frac{1}{\sqrt{t+1}})$, i.e,*

$$\min_{1\le t\le T} g(\boldsymbol{X}_t) \le \frac{2\max\{\triangle_0, \sqrt{L\triangle_0/n}\}}{\sqrt{T+1}}, \qquad (19)$$

*where $\triangle_0 = F_\alpha(\boldsymbol{X}_0) - \min_{X\in\mathcal{D}_n} F_\alpha(\boldsymbol{X})$, and $L$ is the largest absolute eigenvalue of $\nabla^2 J_\alpha(\boldsymbol{X})$. If $J_\alpha(\boldsymbol{X})$ is convex, which happens when $\alpha$ is small (see (8)), then we have a tighter bound $O(\frac{1}{T+1})$.*

**Theorem 2.** *If $\boldsymbol{J}_\alpha(\boldsymbol{X})$ is convex, we have $F_\alpha(\boldsymbol{X}_T) - F_\alpha(\boldsymbol{X}^*) \le \frac{4L}{n(T+1)}$.*

Note that in both cases, convex and non-convex, our EnFW achieves the same (up to a constant coefficient) convergence rate with the original FW algorithm (see [17] and [32]). Thanks to the efficiency of the Sinkhorn-Knopp algorithm, we need much less time to finish each iteration. Therefore, our optimization algorithm is more computationally efficient than the original FW algorithm.

# 6 Experiments

In this section, we conduct extensive experiments to demonstrate the matching performance and scalability of our kernelized graph matching framework. We implement all the algorithms using Matlab on an Intel i7-7820HQ, 2.90 GHz CPU with 64 GB RAM.

**Notations:** We use $\text{KerGM}_\text{I}$ to denote our algorithm when we use exact edge affinity kernels, and use $\text{KerGM}_\text{II}$ to denote it when we use approximate explicit feature maps.

**Baseline methods:** We compare our algorithm with many state-of-the-art graph (network) matching algorithms: (i) Integer projected fixed point method (IPFP) [25], (ii) Spectral matching with affine constraints (SMAC) [9], (iii) Probabilistic graph matching (PM) [46] , (iv) Re-weighted random walk matching (RRWM) [5], (v) Factorized graph matching (FGM) [48], (vi) Branch path following for graph matching (BPFG) [39], (vii) Graduated assignment graph matching (GAGM) [14], (viii) Global network alignment using multiscale spectral signatures (GHOST) [31], (ix) Triangle alignment (TAME) [30], and (x) Maximizing accuracy in global network alignment (MAGNA) [35]. Note that GHOST, TAME, and MAGNA are popular protein-protein interaction (PPI) networks aligners.

**Settings:** For all the baseline methods, we used the parameters recommended in the public code. For our method, if not specified, we set the regularization parameter (see (15)) $\lambda = 0.005$ and the path following parameters $\alpha = 0 : 0.1 : 1$. We use the Hungarian algorithm for final discretization. We refer to the supplementary material for other implementation details.

## 6.1 Synthetic datasets

We evaluate algorithms on the synthetic Erdos–Rényi [12] random graphs, following the experimental protocol in [14, 48, 5]. For each trial, we generate two graphs: the reference graph $\mathcal{G}_1$ and the perturbed graph $\mathcal{G}_2$, each of which has $n_\text{in}$ inlier nodes and $n_\text{out}$ outlier nodes. Each edge in $\mathcal{G}_1$ is randomly generated with probability $\rho \in [0, 1]$. The edges $e_{ij}^1 \in \mathcal{E}_1$ are associated with the edge attributes $q_{ij}^1 \sim \mathcal{U}[0, 1]$. The corresponding edge $e_{p(i)p(j)}^2 \in \mathcal{E}_2$ has the attribute $q_{p(i)p(j)}^2 = q_{ij}^1 + \epsilon$, where $p$ is a permutation map for inlier nodes, and $\epsilon \sim N(0, \sigma^2)$ is the Gaussian noise. For the baseline methods, the edge affinity value between $q_{ij}^1$ and $q_{ij}^2$ is computed as $k^E(q_{ij}^1, q_{ij}^2) = \exp(-(q_{ij}^1 - q_{ij}^2)^2/0.15)$. For our method, we use the Fourier random features (11) to approximate the Gaussian kernel, and represent each graph by an $(n_\text{in} + n_\text{out}) \times (n_\text{in} + n_\text{out})$ array in $\mathbb{R}^D$. We set the parameter $\gamma = 5$ and the dimension $D = 20$.

**Comparing matching accuracy.** We perform the comparison under three parameter settings, in all of which we set $n_\text{in} = 50$. Note that different from the standard protocol where $n_\text{in} = 20$ [48], we use relatively large graphs to highlight the advantage of our $\text{KerGM}_\text{II}$. (i) We change the number of outlier nodes, $n_\text{out}$, from 0 to 50 while fixing the noise, $\sigma = 0$, and the edge density, $\rho = 1$. (ii) We change $\sigma$ from 0 to 0.2 while fixing $n_\text{out} = 0$ and $\rho = 1$. (iii) We change $\rho$ from 0.3 to 1 while fixing $n_\text{out} = 5$ and $\sigma = 0.1$. For all cases in these settings, we repeat the experiments 100 times and report the average accuracy and standard error in Fig. 3 (a). Clearly, our $\text{KerGM}_\text{II}$ outpeforms all the baseline methods with statistical significance.

**Comparing scalability.** To fairly compare the scalability of different algorithms, we consider the exact matching between fully connected graphs, i.e., $n_\text{out} = 0$, $\sigma = 0$, and $\rho = 1$. We change the number of nodes, $n$ ($= n_\text{in}$), from 50 to 2000, and report the CPU time of each algorithm in Fig. 3 (b). We can see that all the baseline methods can handle only graphs with fewer than 200 nodes because of the expensive space cost of matrix $\boldsymbol{K}$ (see (3)). However, $\text{KerGM}_\text{II}$ can finish Lawler's graph matching problem with 2000 nodes in reasonable time.

**Analyzing parameter sensitivity.** To analyze the parameter sensitivity of $\text{KerGM}_\text{II}$, we vary the regularization parameter, $\lambda$, and the dimension, $D$, of Fourier random features. We conduct large subgraph matching experiments by setting $n_\text{in} = 500$, $n_\text{out} = 0 : 100 : 500$, $\rho = 1$, and $\sigma = 0$. We repeat the experiments 50 times and report the average accuracies and standard errors. In Fig. 4, we show the results under different $\lambda$ and different $D$. We can see that (i) smaller $\lambda$ leads to better performance, which can be easily understood because the entropy regularizer will perturb the original optimal solution, and (ii) the dimension $D$ does not much affect on $\text{KerGM}_\text{II}$, which implies that in practice, we can use relatively small $D$ for reducing the time and space complexity.

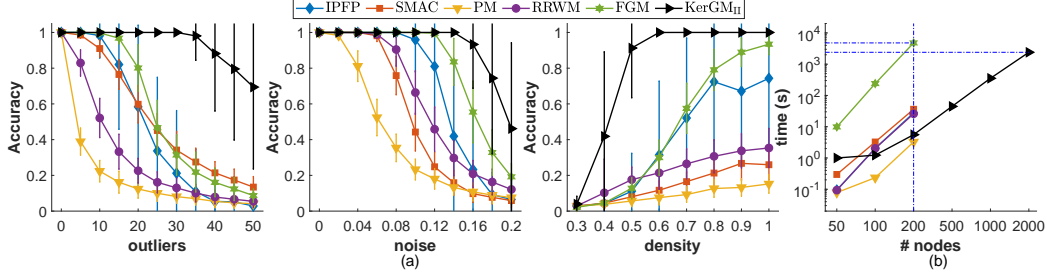

Figure 3: Comparison of graph matching on synthetic datasets.

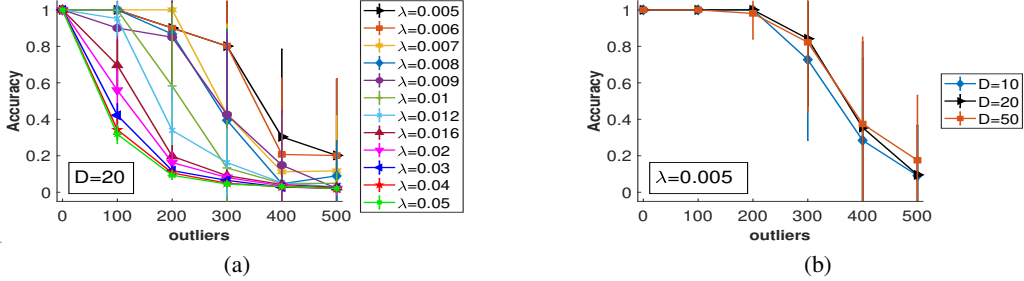

Figure 4: (a) Parameter sensitivity study of the regularizer $\lambda$. (b) Parameter sensitivity study of the dimension, $D$, of the random Fourier feature.

## 6.2 Image datasets

The **CMU House Sequence** dataset has 111 frames of a house, each of which has 30 labeled landmarks. We follow the experimental protocol in [48, 39]. We match all the image pairs, spaced by 0:10:90 frames. We consider two node settings: $(n_1, n_2) = (30, 30)$ and $(n_1, n_2) = (20, 30)$. We build graphs by using Delaunay triangulation [23] to connect landmarks. The edge attributes are the pairwise distances between nodes. For all methods, we compute the edge affinity as $k^E(q_{ij}^1, q_{ab}^2) = \exp(-(q_{ij}^1 - q_{ab}^2)^2/2500)$. In Fig. 5, we report the average matching accuracy and objective function (3) value ratio for every gap. It can be seen that on this dataset, $\mathrm{KerGM_I}$ and FGM achieve the best performance, and are slightly better than BPFG when outliers exist, i.e., $(n_1, n_2) = (20, 30)$.

The **Pascal** dataset [26] has 20 pairs of motorbike images and 30 pairs of car images. For each pair, the detected feature points and manually labeled correspondences are provided. Following [48, 39], we randomly select 0:2:20 outliers from the background to compare different methods. For each node, $v_i$, its attribute, $p_i$, is assigned as its orientation of the normal vector at that point to the contour where the point was sampled. Nodes are connected by Delaunay triangulation [23]. For each edge, $e_{ij}$, its attribute $\vec{q}_{ij}$ equals $[d_{ij}, \theta_{ij}]^T$, where $d_{ij}$ is the distance between $v_i$ and $v_j$, and $\theta_{ij}$ is the absolute angle between the edge and the horizontal line. For all methods, the node affinity is computed as $k^N(p_i, p_j) = \exp(-|p_i - p_j|)$. The edge affinity is computed as $k^E(\vec{q}_{ij}^1, \vec{q}_{ab}^2) = \exp(-|d_{ij}^1 - d_{ab}^2|/2 - |\theta_{ij}^1 - \theta_{ab}^2|/2)$. Fig. 6 (a) shows a matching result of $\mathrm{KerGM_I}$.

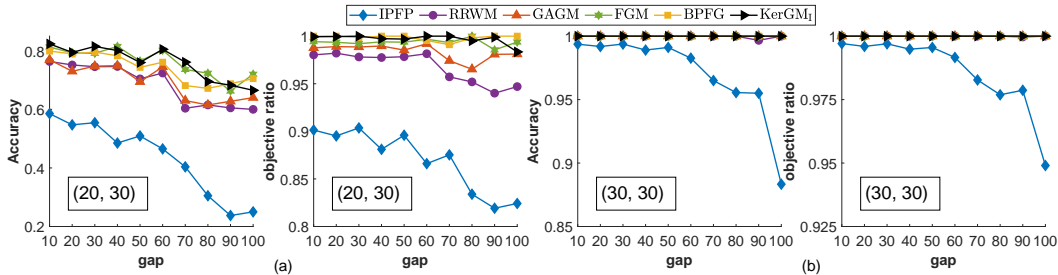

Figure 5: Comparison of graph matching on the CMU house dataset.

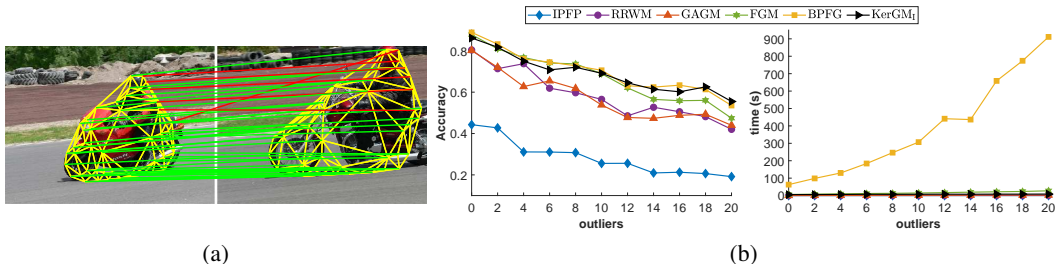

(a)                      (b)

Figure 6: (a) A matching example for a pair of motorbike images generated by $\mathrm{KerGM_I}$, where green and red lines respectively indicate correct and incorrect matches. (b) Comparison of graph matching on the Pascal dataset.

In Fig. 6 (b), we report the matching accuracies and CPU running time. From the perspective of matching accuracy, $\mathrm{KerGM_I}$, BPFG, and FGM consistently outperforms other methods. When the number of outliers increases, $\mathrm{KerGM_I}$ and BPFG perform slightly better than FGM. However, from the perspective of running time, the time cost of BPFG is much higher than that of the others.

### 6.3 The protein-protein interaction network dataset

The **S.cerevisiae (yeast) PPI network** [7] dataset is popularly used to evaluate PPI network aligners because it has known true node correspondences.

It consists of an unweighted high-confidence PPI network with 1004 proteins (nodes) and 8323 PPIs (edges), and five noisy PPI networks generated by adding 5%, 10%, 15%, 20%, 25% low-confidence PPIs. We do graph matching between the high-confidence network with every noisy network. To apply KerGM, we generate edge attributes by the heat diffusion matrix [16, 6], $\boldsymbol{H}_t = \exp(-t\boldsymbol{L}) = \sum_{i=1}^{n} \exp(-\lambda_i t)\vec{\boldsymbol{u}}_i\vec{\boldsymbol{u}}_i^T \in \mathbb{R}^{n \times n}$, where $\boldsymbol{L}$ is the normalized Laplacian matrix [6], and $\{(\lambda_i, \vec{\boldsymbol{u}}_i)\}_{i=1}^{n}$ are eigenpairs of $\boldsymbol{L}$. The edge attributes vector $\vec{\boldsymbol{q}}_{ij}$ is assigned as $\vec{\boldsymbol{q}}_{ij} = [\boldsymbol{H}_5(i,j), \boldsymbol{H}_{10}(i,j), \boldsymbol{H}_{15}(i,j), \boldsymbol{H}_{20}(i,j)]^T \in$

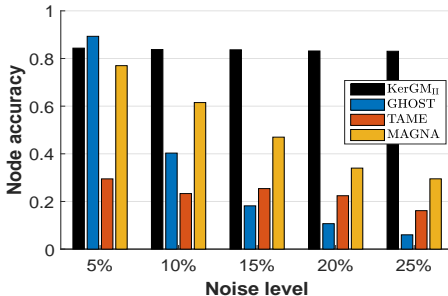

Figure 7: Results on PPI networks.

$\mathbb{R}^4$. We use the Fourier random features (11), and set $D = 50$ and $\gamma = 200$. We compare $\mathrm{KerGM_{II}}$[3] with the state-of-the-art PPI aligners: TAME, GHOST, and MAGNA. In Fig. 7, we report the matching accuracies. Clearly, $\mathrm{KerGM_{II}}$ significantly outperforms the baselines. Especially when the noise level are 20% or 25%, $\mathrm{KerGM_{II}}$'s accuracies are more than 50 percentages higher than those of other algorithms.

## 7 Conclusion

In this work, based on a mild assumption regarding edge affinity values, we provided KerGM, a unifying framework for Koopman-Beckmann's and Lawler's QAPs, within which both two QAPs can be considered as the alignment between arrays in RKHS. Then we derived convex and concave relaxations and the corresponding path-following strategy. To make KerGM more scalable to large graphs, we developed the computationally efficient entropy-regularized Frank-Wolfe optimization algorithm. KerGM achieved promising performance on both image and biology datasets. Thanks to its scalability, we believe KerGM can be potentially useful for many applications in the real world.

## 8 Acknowledgment

This work was supported in part by the AFOSR grant FA9550-16-1-0386.

## Footnotes

[1] We assume $\mathcal{G}_1$ and $\mathcal{G}_2$ have the same number of nodes. If not, we add dummy nodes.

[2]For convenience in developing the path-following strategy, we write it in the minimization form.

[3]To the best our knowledge, KerGM is the first one that uses Lawler's graph matching formulation to solve the PPI network alignment problem.

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
