[Supplementary Material]

# KerGM: Kernelized Graph Matching: Supplementary Material

**Zhen Zhang[1], Yijian Xiang[1], Lingfei Wu[2], Bing Xue[1], Arye Nehorai[1]**
[1]Washington University in St. Louis
[2]IBM Research
[1]{zhen.zhang, yijian.xiang, xuebing, nehorai}@wustl.edu
[2]lwu@email.wm.edu

## Abstract

The supplementary material consists of three parts. In the first part, we prove all the mathematical results of the kernelized graph matching formulation. We also provide additional technical discussions. In the second part, we prove all the mathematical results of the entropy-regularized Frank-Wolfe algorithm. In the third part, we describe the implementation details and give more experimental results.

## 1 Proofs of the mathematical results of kernelized graph matching

### 1.1 Proving Corollary 1

**Corollary 1.** $\forall \boldsymbol{X}, \boldsymbol{Y} \in \mathbb{R}^{n \times n}$, $\boldsymbol{\Psi} \odot \boldsymbol{X} \odot \boldsymbol{Y} = \boldsymbol{\Psi} \odot (\boldsymbol{X}\boldsymbol{Y})$, and $\boldsymbol{Y} \odot (\boldsymbol{X} \odot \boldsymbol{\Psi}) = (\boldsymbol{Y}\boldsymbol{X}) \odot \boldsymbol{\Psi}$.

*Proof.*
**(1).**
$\forall i, j = 1, 2, ..., n$, the $(i, j)$ element of $\boldsymbol{\Psi} \odot \boldsymbol{X} \odot \boldsymbol{Y}$ is

$$
\begin{aligned}
[\boldsymbol{\Psi} \odot \boldsymbol{X} \odot \boldsymbol{Y}]_{ij} &= \sum_{k=1}^{n} [\boldsymbol{\Psi}\boldsymbol{X}]_{ik} \boldsymbol{Y}_{kj} = \sum_{k=1}^{n} \boldsymbol{Y}_{kj} \sum_{\alpha=1}^{n} \boldsymbol{\Psi}_{i\alpha} \boldsymbol{X}_{\alpha k} \\
&= \sum_{\alpha=1}^{n} \boldsymbol{\Psi}_{i\alpha} \big( \sum_{k=1}^{n} \boldsymbol{X}_{\alpha k} \boldsymbol{Y}_{kj} \big) = \sum_{\alpha=1}^{n} \boldsymbol{\Psi}_{i\alpha} [\boldsymbol{X}\boldsymbol{Y}]_{\alpha j} = [\boldsymbol{\Psi} \odot (\boldsymbol{X}\boldsymbol{Y})]_{ij}.
\end{aligned}
\tag{1}
$$

Therefore, $\boldsymbol{\Psi} \odot \boldsymbol{X} \odot \boldsymbol{Y} = \boldsymbol{\Psi} \odot (\boldsymbol{X}\boldsymbol{Y})$.

**(2).**
$\forall i, j = 1, 2, ..., n$, the $(i, j)$ element of $\boldsymbol{Y} \odot (\boldsymbol{X} \odot \boldsymbol{\Psi})$ is

$$
[\boldsymbol{Y} \odot (\boldsymbol{X} \odot \boldsymbol{\Psi})]_{ij} = \sum_{k=1}^{n} \boldsymbol{Y}_{ik} [\boldsymbol{X}\boldsymbol{\Psi}]_{kj} = \sum_{k=1}^{n} \boldsymbol{Y}_{ik} \sum_{\alpha=1}^{n} \boldsymbol{X}_{k\alpha} \boldsymbol{\Psi}_{\alpha j}
\tag{2}
$$

$$
= \sum_{\alpha=1}^{n} \big( \sum_{k=1}^{n} \boldsymbol{Y}_{ik} \boldsymbol{X}_{k\alpha} \big) \boldsymbol{\Psi}_{\alpha j} = \sum_{\alpha=1}^{n} [\boldsymbol{Y}\boldsymbol{X}]_{i\alpha} \boldsymbol{\Psi}_{\alpha j} = [(\boldsymbol{Y}\boldsymbol{X}) \odot \boldsymbol{\Psi}]_{ij}.
\tag{3}
$$

Therefore, $\boldsymbol{Y} \odot (\boldsymbol{X} \odot \boldsymbol{\Psi}) = (\boldsymbol{Y}\boldsymbol{X}) \odot \boldsymbol{\Psi}$. □

## 1.2 Proving Proposition 1

**Proposition 1.** *Define the function* $\langle \cdot, \cdot \rangle_{\mathrm{F}_{\mathcal{H}}} : \mathcal{H}^{n \times n} \times \mathcal{H}^{n \times n} \to \mathbb{R}$ *such that* $\langle \boldsymbol{\Psi}, \boldsymbol{\Xi} \rangle_{\mathrm{F}_{\mathcal{H}}} \triangleq \mathrm{tr}(\boldsymbol{\Psi}^T *$ $\boldsymbol{\Xi}) = \sum_{i,j=1}^{n} \langle \boldsymbol{\Psi}_{ij}, \boldsymbol{\Xi}_{ij} \rangle_{\mathcal{H}}$, $\forall \boldsymbol{\Psi}, \boldsymbol{\Xi} \in \mathcal{H}^{n \times n}$. *Then* $\langle \cdot, \cdot \rangle_{\mathrm{F}_{\mathcal{H}}}$ *induces an inner product on* $\mathcal{H}^{n \times n}$.

*Proof.* It is sufficient to show that the function $\langle \cdot, \cdot \rangle_{\mathrm{F}_{\mathcal{H}}}$ satisfies the following properties.

**1. [Conjugate symmetry]:**

$$\overline{\langle \boldsymbol{\Psi}, \boldsymbol{\Xi} \rangle_{\mathrm{F}_{\mathcal{H}}}} = \sum_{i,j=1}^{n} \overline{\langle \boldsymbol{\Psi}_{ij}, \boldsymbol{\Xi}_{ij} \rangle_{\mathcal{H}}} = \sum_{i,j=1}^{n} \langle \boldsymbol{\Xi}_{ij}, \boldsymbol{\Psi}_{ij} \rangle_{\mathcal{H}} = \langle \boldsymbol{\Xi}, \boldsymbol{\Psi} \rangle_{\mathrm{F}_{\mathcal{H}}}, \tag{4}$$

**2. [Linearity in the first argument]:**

$$\langle a\boldsymbol{\Psi}, \boldsymbol{\Xi} \rangle_{\mathrm{F}_{\mathcal{H}}} = \sum_{i,j=1}^{n} \langle a\boldsymbol{\Psi}_{ij}, \boldsymbol{\Xi}_{ij} \rangle_{\mathcal{H}} = \sum_{i,j=1}^{n} a\langle \boldsymbol{\Psi}_{ij}, \boldsymbol{\Xi}_{ij} \rangle_{\mathcal{H}} = a\langle \boldsymbol{\Psi}, \boldsymbol{\Xi} \rangle_{\mathrm{F}_{\mathcal{H}}} \tag{5}$$

$$\langle \boldsymbol{\Psi}^{(1)} + \boldsymbol{\Psi}^{(2)}, \boldsymbol{\Xi} \rangle_{\mathrm{F}_{\mathcal{H}}} = \sum_{i,j=1}^{n} \langle \boldsymbol{\Psi}_{ij}^{(1)} + \boldsymbol{\Psi}_{ij}^{(2)}, \boldsymbol{\Xi}_{ij} \rangle_{\mathcal{H}} = \sum_{i,j=1}^{n} \langle \boldsymbol{\Psi}_{ij}^{(1)}, \boldsymbol{\Xi}_{ij} \rangle_{\mathcal{H}} + \sum_{i,j=1}^{n} \langle \boldsymbol{\Psi}_{ij}^{(2)}, \boldsymbol{\Xi}_{ij} \rangle_{\mathcal{H}}. \tag{6}$$

$$= \langle \boldsymbol{\Psi}^{(1)}, \boldsymbol{\Xi} \rangle_{\mathrm{F}_{\mathcal{H}}} + \langle \boldsymbol{\Psi}^{(2)}, \boldsymbol{\Xi} \rangle_{\mathrm{F}_{\mathcal{H}}} \tag{7}$$

**3. [Positive-definiteness]:**

$$\langle \boldsymbol{\Psi}, \boldsymbol{\Psi} \rangle_{\mathrm{F}_{\mathcal{H}}} = \sum_{i,j=1}^{n} \langle \boldsymbol{\Psi}_{ij}, \boldsymbol{\Psi}_{ij} \rangle_{\mathcal{H}} \geq 0. \tag{8}$$

$$\langle \boldsymbol{\Psi}, \boldsymbol{\Psi} \rangle_{\mathrm{F}_{\mathcal{H}}} = 0 \iff \forall i, j = 1, 2, ..., n, \boldsymbol{\Psi}_{ij} = \boldsymbol{0} \iff \boldsymbol{\Psi} = \boldsymbol{O}. \tag{9}$$

□

## 1.3 Proving Corollary 2

**Corollary 2.** $\langle \boldsymbol{\Psi} \odot \boldsymbol{X}, \boldsymbol{\Xi} \rangle_{\mathrm{F}_{\mathcal{H}}} = \langle \boldsymbol{\Psi}, \boldsymbol{\Xi} \odot \boldsymbol{X}^T \rangle_{\mathrm{F}_{\mathcal{H}}}$ *and* $\langle \boldsymbol{X} \odot \boldsymbol{\Psi}, \boldsymbol{\Xi} \rangle_{\mathrm{F}_{\mathcal{H}}} = \langle \boldsymbol{\Psi}, \boldsymbol{X}^T \odot \boldsymbol{\Xi} \rangle_{\mathrm{F}_{\mathcal{H}}}$.

*Proof.*
**(1).**

$$\langle \boldsymbol{\Psi} \odot \boldsymbol{X}, \boldsymbol{\Xi} \rangle_{\mathrm{F}_{\mathcal{H}}} = \sum_{i=1}^{n} \sum_{j=1}^{n} \langle [\boldsymbol{\Psi} \odot \boldsymbol{X}]_{ij}, \boldsymbol{\Xi}_{ij} \rangle_{\mathcal{H}} = \sum_{i=1}^{n} \sum_{j=1}^{n} \langle \sum_{k=1}^{n} \boldsymbol{\Psi}_{ik} \boldsymbol{X}_{kj}, \boldsymbol{\Xi}_{ij} \rangle_{\mathcal{H}}$$

$$= \sum_{i=1}^{n} \sum_{j=1}^{n} \sum_{k=1}^{n} \langle \boldsymbol{\Psi}_{ik}, \boldsymbol{X}_{kj} \boldsymbol{\Xi}_{ij} \rangle_{\mathcal{H}} = \sum_{i=1}^{n} \sum_{k=1}^{n} \langle \boldsymbol{\Psi}_{ik}, \sum_{j=1}^{n} \boldsymbol{X}_{kj} \boldsymbol{\Xi}_{ij} \rangle_{\mathcal{H}}$$

$$= \sum_{i=1}^{n} \sum_{k=1}^{n} \langle \boldsymbol{\Psi}_{ik}, [\boldsymbol{\Xi} \odot \boldsymbol{X}^T]_{ik} \rangle_{\mathcal{H}} = \langle \boldsymbol{\Psi}, \boldsymbol{\Xi} \odot \boldsymbol{X}^T \rangle_{\mathrm{F}_{\mathcal{H}}}. \tag{10}$$

**(2).**

$$\langle \boldsymbol{X} \odot \boldsymbol{\Psi}, \boldsymbol{\Xi} \rangle_{\mathrm{F}_{\mathcal{H}}} = \sum_{i=1}^{n} \sum_{j=1}^{n} \langle [\boldsymbol{X} \odot \boldsymbol{\Psi}]_{ij}, \boldsymbol{\Xi}_{ij} \rangle_{\mathcal{H}} = \sum_{i=1}^{n} \sum_{j=1}^{n} \langle \sum_{k=1}^{n} \boldsymbol{X}_{ik} \boldsymbol{\Psi}_{kj}, \boldsymbol{\Xi}_{ij} \rangle_{\mathcal{H}} \tag{11}$$

$$= \sum_{i=1}^{n} \sum_{j=1}^{n} \sum_{k=1}^{n} \langle \boldsymbol{\Psi}_{kj}, \boldsymbol{X}_{ik} \boldsymbol{\Xi}_{ij} \rangle_{\mathcal{H}} = \sum_{j=1}^{n} \sum_{k=1}^{n} \langle \boldsymbol{\Psi}_{kj}, \sum_{i=1}^{n} \boldsymbol{X}_{ik} \boldsymbol{\Xi}_{ij} \rangle_{\mathcal{H}} \tag{12}$$

$$= \sum_{j=1}^{n} \sum_{k=1}^{n} \langle \boldsymbol{\Psi}_{kj}, [\boldsymbol{X}^T \odot \boldsymbol{\Xi}]_{kj} \rangle_{\mathcal{H}} = \langle \boldsymbol{\Psi}, \boldsymbol{X}^T \odot \boldsymbol{\Xi} \rangle_{\mathrm{F}_{\mathcal{H}}}. \tag{13}$$

□

## 1.4 Computing the gradient

$$\nabla J_\alpha(\boldsymbol{X}) = (1 - 2\alpha)\big[(\boldsymbol{\Psi}^{(1)} * \boldsymbol{\Psi}^{(1)})\boldsymbol{X} + \boldsymbol{X}(\boldsymbol{\Psi}^{(2)} * \boldsymbol{\Psi}^{(2)})\big] - 2(\boldsymbol{\Psi}^{(1)} \odot \boldsymbol{X}) * \boldsymbol{\Psi}^{(2)} - \boldsymbol{K}^N, \quad (14)$$

where $\forall i,j = 1,2,...,n$, $[\boldsymbol{\Psi}^{(1)} * \boldsymbol{\Psi}^{(1)}]_{ij} = \sum_{e_{ik}^1, e_{kj}^1 \in \mathcal{E}_1} k^E(\vec{q}_{ik}^1, \vec{q}_{kj}^1)$; $\forall a,b = 1,2,...,n$, $[\boldsymbol{\Psi}^{(2)} * \boldsymbol{\Psi}^{(2)}]_{ab} = \sum_{e_{ac}^2, e_{cb}^2 \in \mathcal{E}_2} k^E(\vec{q}_{ac}^2, \vec{q}_{cb}^2)$; and $\forall i,a = 1,2,...,n$, $[(\boldsymbol{\Psi}^{(1)} \odot \boldsymbol{X}) * \boldsymbol{\Psi}^{(2)}]_{ia} = \sum_{e_{ik}^1 \in \mathcal{E}_1, e_{ca}^2 \in \mathcal{E}_2} \boldsymbol{X}_{kc} k^E(\vec{q}_{ik}^1, \vec{q}_{ca}^2)$.

**We first present two useful lemmas.**

**Lemma 1.** $\langle \boldsymbol{\Psi}, \boldsymbol{\Xi} \odot \boldsymbol{X} \rangle_{\mathrm{F}_\mathcal{H}} = \langle \boldsymbol{\Xi}^T * \boldsymbol{\Psi}, \boldsymbol{X} \rangle_{\mathrm{F}}$, and $\langle \boldsymbol{\Psi}, \boldsymbol{X} \odot \boldsymbol{\Xi} \rangle_{\mathrm{F}_\mathcal{H}} = \langle \boldsymbol{\Psi} * \boldsymbol{\Xi}^T, \boldsymbol{X} \rangle_{\mathrm{F}}$.

**Remark 1.** *In our paper, we only consider the kernel values. That is, the inner product values are real numbers. So* $\langle \psi, \varphi \rangle_\mathcal{H} = \langle \varphi, \psi \rangle_\mathcal{H}, \forall \psi, \varphi \in \mathcal{H}$.

*Proof.*
**(1).**

$$\langle \boldsymbol{\Psi}, \boldsymbol{\Xi} \odot \boldsymbol{X} \rangle_{\mathrm{F}_\mathcal{H}} = \sum_{i=1}^n \sum_{j=1}^n \langle \boldsymbol{\Psi}_{ij}, [\boldsymbol{\Xi} \odot \boldsymbol{X}]_{ij} \rangle_\mathcal{H} = \sum_{i=1}^n \sum_{j=1}^n \langle \boldsymbol{\Psi}_{ij}, \sum_{k=1}^n \boldsymbol{\Xi}_{ik} \boldsymbol{X}_{kj} \rangle_\mathcal{H}$$

$$= \sum_{i=1}^n \sum_{j=1}^n \sum_{k=1}^n \langle \boldsymbol{\Psi}_{ij}, \boldsymbol{\Xi}_{ik} \boldsymbol{X}_{kj} \rangle_\mathcal{H} = \sum_{k=1}^n \sum_{j=1}^n \boldsymbol{X}_{kj} \sum_{i=1}^n \langle \boldsymbol{\Psi}_{ij}, \boldsymbol{\Xi}_{ik} \rangle_\mathcal{H}$$

$$= \sum_{k=1}^n \sum_{j=1}^n [\boldsymbol{\Xi}^T * \boldsymbol{\Psi}]_{kj} \boldsymbol{X}_{kj} = \langle \boldsymbol{\Xi}^T * \boldsymbol{\Psi}, \boldsymbol{X} \rangle_{\mathrm{F}}. \quad (15)$$

**(2).**

$$\langle \boldsymbol{\Psi}, \boldsymbol{X} \odot \boldsymbol{\Xi} \rangle_{\mathrm{F}_\mathcal{H}} = \sum_{i=1}^n \sum_{j=1}^n \langle \boldsymbol{\Psi}_{ij}, [\boldsymbol{X} \odot \boldsymbol{\Xi}]_{ij} \rangle_\mathcal{H} = \sum_{i=1}^n \sum_{j=1}^n \langle \boldsymbol{\Psi}_{ij}, \sum_{k=1}^n \boldsymbol{X}_{ik} \boldsymbol{\Xi}_{kj} \rangle_\mathcal{H}$$

$$= \sum_{i=1}^n \sum_{j=1}^n \sum_{k=1}^n \langle \boldsymbol{\Psi}_{ij}, \boldsymbol{X}_{ik} \boldsymbol{\Xi}_{kj} \rangle_\mathcal{H} = \sum_{i=1}^n \sum_{k=1}^n \boldsymbol{X}_{ik} \sum_{j=1}^n \langle \boldsymbol{\Psi}_{ij}, \boldsymbol{\Xi}_{kj} \rangle_\mathcal{H}$$

$$= \sum_{i=1}^n \sum_{k=1}^n [\boldsymbol{\Psi} * \boldsymbol{\Xi}^T]_{ik} \boldsymbol{X}_{ik} = \langle \boldsymbol{\Psi} * \boldsymbol{\Xi}^T, \boldsymbol{X} \rangle_{\mathrm{F}}. \quad (16)$$

$\square$

**Lemma 2.** $\boldsymbol{\Psi} * (\boldsymbol{\Xi} \odot \boldsymbol{X}) = (\boldsymbol{\Psi} * \boldsymbol{\Xi})\boldsymbol{X}$, $(\boldsymbol{X} \odot \boldsymbol{\Psi}) * \boldsymbol{\Xi} = \boldsymbol{X}(\boldsymbol{\Psi} * \boldsymbol{\Xi})$, *and* $\boldsymbol{\Psi} * (\boldsymbol{X} \odot \boldsymbol{\Xi}) = (\boldsymbol{\Psi} \odot \boldsymbol{X}) * \boldsymbol{\Xi}$.

*Proof.* The proof procedure is very similar with Corollary 1. $\square$

**Now we prove the equality** (14)**.**

*Proof.*
**(1).**
We first rewrite the function $J_\alpha(\boldsymbol{X})$ as

$$J_\alpha(\boldsymbol{X}) = -\langle \boldsymbol{K}^N, \boldsymbol{X} \rangle_{\mathrm{F}} + (1 - \alpha)J_1(\boldsymbol{X}) - \alpha J_2(\boldsymbol{X}),$$

where $J_1(\boldsymbol{X}) = \frac{1}{2} \|\boldsymbol{\Psi}^{(1)} \odot \boldsymbol{X} - \boldsymbol{X} \odot \boldsymbol{\Psi}^{(2)}\|_{\mathrm{F}_\mathcal{H}}^2$ and $J_2(\boldsymbol{X}) = \frac{1}{2} \|\boldsymbol{\Psi}^{(1)} \odot \boldsymbol{X} + \boldsymbol{X} \odot \boldsymbol{\Psi}^{(2)}\|_{\mathrm{F}_\mathcal{H}}^2$

We first compute the gradient of $J_1(\boldsymbol{X})$. We employ the following fact:

$$\forall \boldsymbol{E} \in \mathbb{R}^{n \times n}, \; \langle \nabla J_1(\boldsymbol{X}), \boldsymbol{E} \rangle_{\mathrm{F}} = \lim_{t \to 0} \frac{J_1(\boldsymbol{X} + t\boldsymbol{E}) - J_1(\boldsymbol{X})}{t}. \quad (17)$$

$$J_1(\boldsymbol{X} + t\boldsymbol{E}) - J_1(\boldsymbol{X})$$

$$= \frac{1}{2}\|\boldsymbol{\Psi}^{(1)} \odot (\boldsymbol{X} + t\boldsymbol{E}) - (\boldsymbol{X} + t\boldsymbol{E}) \odot \boldsymbol{\Psi}^{(2)}\|^2_{\mathrm{F}_{\mathcal{H}}} - \frac{1}{2}\|\boldsymbol{\Psi}^{(1)} \odot \boldsymbol{X} - \boldsymbol{X} \odot \boldsymbol{\Psi}^2\|^{(2)}_{\mathrm{F}_{\mathcal{H}}} \qquad (18)$$

$$= \frac{1}{2}t^2\|\boldsymbol{\Psi}^{(1)} \odot \boldsymbol{E} - \boldsymbol{E} \odot \boldsymbol{\Psi}^{(2)}\|^2_{\mathrm{F}_{\mathcal{H}}} + t\langle \boldsymbol{\Psi}^{(1)} \odot \boldsymbol{X} - \boldsymbol{X} \odot \boldsymbol{\Psi}^{(2)}, \boldsymbol{\Psi}^{(1)} \odot \boldsymbol{E} - \boldsymbol{E} \odot \boldsymbol{\Psi}^{(2)}\rangle_{\mathrm{F}_{\mathcal{H}}}$$

Immediately, $\langle \nabla J_1(\boldsymbol{X}), \boldsymbol{E}\rangle_{\mathrm{F}} = \langle \boldsymbol{\Psi}^{(1)} \odot \boldsymbol{X} - \boldsymbol{X} \odot \boldsymbol{\Psi}^{(2)}, \boldsymbol{\Psi}^{(1)} \odot \boldsymbol{E} - \boldsymbol{E} \odot \boldsymbol{\Psi}^{(2)}\rangle_{\mathrm{F}_{\mathcal{H}}}$.

We can rewrite the above formula as

$$\langle \boldsymbol{\Psi}^{(1)} \odot \boldsymbol{X} - \boldsymbol{X} \odot \boldsymbol{\Psi}^{(2)}, \boldsymbol{\Psi}^{(1)} \odot \boldsymbol{E} - \boldsymbol{E} \odot \boldsymbol{\Psi}^{(2)}\rangle_{\mathrm{F}_{\mathcal{H}}}$$

$$= \langle \boldsymbol{\Psi}^{(1)} \odot \boldsymbol{X} - \boldsymbol{X} \odot \boldsymbol{\Psi}^{(2)}, \boldsymbol{\Psi}^{(1)} \odot \boldsymbol{E}\rangle_{\mathrm{F}_{\mathcal{H}}} - \langle \boldsymbol{\Psi}^{(1)} \odot \boldsymbol{X} - \boldsymbol{X} \odot \boldsymbol{\Psi}^{(2)}, \boldsymbol{E} \odot \boldsymbol{\Psi}^{(2)}\rangle_{\mathrm{F}_{\mathcal{H}}}$$

$$= \langle \boldsymbol{\Psi}^{(1)} * (\boldsymbol{\Psi}^{(1)} \odot \boldsymbol{X}) - \boldsymbol{\Psi}^{(1)} * (\boldsymbol{X} \odot \boldsymbol{\Psi}^{(2)}), \boldsymbol{E}\rangle_{\mathrm{F}} - \langle (\boldsymbol{\Psi}^{(1)} \odot \boldsymbol{X}) * \boldsymbol{\Psi}^{(2)} - (\boldsymbol{X} \odot \boldsymbol{\Psi}^{(2)}) * \boldsymbol{\Psi}^{(2)}, \boldsymbol{E}\rangle_{\mathrm{F}_{\mathcal{H}}}$$

$$= \langle (\boldsymbol{\Psi}^{(1)} * \boldsymbol{\Psi}^{(1)})\boldsymbol{X} - 2(\boldsymbol{\Psi}^{(1)} \odot \boldsymbol{X}) * \boldsymbol{\Psi}^{(2)} + \boldsymbol{X}(\boldsymbol{\Psi}^{(2)} * \boldsymbol{\Psi}^{(2)}), \boldsymbol{E}\rangle_{\mathrm{F}},$$

$$(19)$$

where the 3rd equality holds because of Lemma 1 and both $\boldsymbol{\Psi}^{(1)}$ and $\boldsymbol{\Psi}^{(2)}$ are symmetric, and the last equality holds because of Lemma 2. Therefore,

$$\nabla J_1(\boldsymbol{X}) = (\boldsymbol{\Psi}^{(1)} * \boldsymbol{\Psi}^{(1)})\boldsymbol{X} - 2(\boldsymbol{\Psi}^{(1)} \odot \boldsymbol{X}) * \boldsymbol{\Psi}^{(2)} + \boldsymbol{X}(\boldsymbol{\Psi}^{(2)} * \boldsymbol{\Psi}^{(2)}).$$

Similarly, we have

$$\nabla J_2(\boldsymbol{X}) = (\boldsymbol{\Psi}^{(1)} * \boldsymbol{\Psi}^{(1)})\boldsymbol{X} + 2(\boldsymbol{\Psi}^{(1)} \odot \boldsymbol{X}) * \boldsymbol{\Psi}^{(2)} + \boldsymbol{X}(\boldsymbol{\Psi}^{(2)} * \boldsymbol{\Psi}^{(2)}).$$

Finally, substituting $\nabla J_1(\boldsymbol{X})$ and $\nabla J_2(\boldsymbol{X})$ into $\nabla J_\alpha(\boldsymbol{X}) = -\boldsymbol{K}^N + (1-\alpha)\nabla J_1(\boldsymbol{X}) - \alpha \nabla J_2(\boldsymbol{X})$, we obtain the result (14).

**(2).**

For the first term $\boldsymbol{\Psi}^{(1)} * \boldsymbol{\Psi}^{(1)}$, we have

$$[\boldsymbol{\Psi}^{(1)} * \boldsymbol{\Psi}^{(1)}]_{ij} = \sum_{k=1}^{n} \langle \boldsymbol{\Psi}^{(1)}_{ik}, \boldsymbol{\Psi}^{(1)}_{kj}\rangle_{\mathcal{H}_{\mathcal{K}}} = \sum_{e^1_{ik}, e^1_{kj} \in \mathcal{E}_1} \langle \psi(\vec{\boldsymbol{q}}^1_{ik}), \psi(\vec{\boldsymbol{q}}^1_{kj})\rangle_{\mathcal{H}_{\mathcal{K}}} = \sum_{e^1_{ik}, e^1_{kj} \in \mathcal{E}_1} k^E(\vec{\boldsymbol{q}}^1_{ik}, \vec{\boldsymbol{q}}^1_{kj}).$$

$$(20)$$

For the second term $\boldsymbol{\Psi}^{(2)} * \boldsymbol{\Psi}^{(2)}$, we have similar explanations.

For the third term $(\boldsymbol{\Psi}^{(1)} \odot \boldsymbol{X}) * \boldsymbol{\Psi}^{(2)}$, we have

$$[(\boldsymbol{\Psi}^{(1)} \odot \boldsymbol{X}) * \boldsymbol{\Psi}^{(2)}]_{ia} = \sum_{c=1}^{n} \langle [\boldsymbol{\Psi}^{(1)} \odot \boldsymbol{X}]_{ic}, \boldsymbol{\Psi}^{(2)}_{ca}\rangle_{\mathcal{H}_{\mathcal{K}}} = \sum_{c=1}^{n} \langle \sum_{k=1}^{n} \boldsymbol{\Psi}^{(1)}_{ik} \boldsymbol{X}_{kc}, \boldsymbol{\Psi}^{(2)}_{ca}\rangle_{\mathcal{H}_{\mathcal{K}}}$$

$$= \sum_{c=1}^{n} \sum_{k=1}^{n} \langle \boldsymbol{\Psi}^{(1)}_{ik} \boldsymbol{X}_{kc}, \boldsymbol{\Psi}^{(2)}_{ca}\rangle_{\mathcal{H}_{\mathcal{K}}} = \sum_{e^1_{ik} \in \mathcal{E}_1, e^2_{ca} \in \mathcal{E}_2} \boldsymbol{X}_{kc} \langle \psi(\vec{\boldsymbol{q}}^1_{ik}), \psi(\vec{\boldsymbol{q}}^2_{ca})\rangle_{\mathcal{H}_{\mathcal{K}}}$$

$$= \sum_{e^1_{ik} \in \mathcal{E}_1, e^2_{ca} \in \mathcal{E}_2} \boldsymbol{X}_{kc} k^E(\vec{\boldsymbol{q}}^1_{ik}, \vec{\boldsymbol{q}}^2_{ca}). \qquad (21)$$

$\square$

### 1.4.1 Gradients in compact matrix multiplication forms

In this section, we rewrite the terms of (14) in compact matrix multiplication forms, providing a convenient way to compute gradients. We first give some necessary definitions.

1. Given a graph $\mathcal{G} = \{\boldsymbol{A}, \mathcal{V}, \boldsymbol{P}, \mathcal{E}, \boldsymbol{Q}\}$ of $n$ nodes and $m$ edges. We define the Head-incidence matrix $\boldsymbol{G} \in \{0,1\}^{n \times m}$ and the Tail-incidence matrix $\boldsymbol{H} \in \{0,1\}^{n \times m}$. For any edge $e_{ij} \in \mathcal{E}$, we arbitrarily assign a direction on $e_{ij}$, e.g., $v_i \to v_j$ or $v_j \to v_i$. Suppose that the artifically assigned direction of $e_{ij}$ is $v_j \to v_i$, then the items $\boldsymbol{G}(v_j, e_{ij}) = 1$ and $\boldsymbol{H}(v_i, e_{ij}) = 1$. A toy example is shown in Fig. 1.

Figure 1: (a) A toy Graph $\mathcal{G}_1$, and its Head-incidence matrix $\boldsymbol{G}_1$ and Tail-incidence matrix $\boldsymbol{H}_1$; (b) A toy Graph $\mathcal{G}_2$, and its Head-incidence matrix $\boldsymbol{G}_2$ and Tail-incidence matrix $\boldsymbol{H}_2$.

2. Given two graphs $\mathcal{G}_1 = \{\boldsymbol{A}_1, \mathcal{V}_1, \boldsymbol{P}_1, \mathcal{E}_1, \boldsymbol{Q}_1\}$ of $n_1$ nodes and $m_1$ edges, and $\mathcal{G}_2 = \{\boldsymbol{A}_2, \mathcal{V}_2, \boldsymbol{P}_2, \mathcal{E}_2, \boldsymbol{Q}_2\}$ of $n_2$ nodes and $m_2$ edges, let $\boldsymbol{K}_{11}^E \in \mathbb{R}^{m_1 \times m_1}$, $\boldsymbol{K}_{22}^E \in \mathbb{R}^{m_2 \times m_2}$, and $\boldsymbol{K}_{12}^E \in \mathbb{R}^{m_1 \times m_2}$ be three kernel matrices induced by the kernel $k^E$ (the edge affinity function). They are defined such that

$$[\boldsymbol{K}_{11}^E](e_{i_1 j_1}^1, e_{i_2 j_2}^1) = k^E(\vec{\boldsymbol{q}}_{i_1 j_1}^1, \vec{\boldsymbol{q}}_{i_2 j_2}^1), \quad \text{if } e_{i_1 j_1}^1, e_{i_2 j_2}^1 \in \mathcal{E}_1, \tag{22}$$

$$[\boldsymbol{K}_{22}^E](e_{a_1 b_1}^2, e_{a_2 b_2}^2) = k^E(\vec{\boldsymbol{q}}_{a_1 b_1}^2, \vec{\boldsymbol{q}}_{a_2 b_2}^2), \quad \text{if } e_{a_1 b_1}^2, e_{a_2 b_2}^2 \in \mathcal{E}_2, \tag{23}$$

$$[\boldsymbol{K}_{12}^E](e_{ij}^1, e_{ab}^2) = k^E(\vec{\boldsymbol{q}}_{ij}^1, \vec{\boldsymbol{q}}_{ab}^2), \quad \text{if } e_{ij}^1 \in \mathcal{E}_1, e_{ab}^2 \in \mathcal{E}_2, \tag{24}$$

Let $\boldsymbol{G}_1$ and $\boldsymbol{H}_1$, and $\boldsymbol{G}_2$ and $\boldsymbol{H}_2$ be the Head-incidence matrix and the Tail-incidence matrix of graph $\mathcal{G}_1$ and $\mathcal{G}_2$ (see Fig. 1), respectively. Then the terms in (14) can be written as

$$\boldsymbol{\Psi}^{(1)} * \boldsymbol{\Psi}^{(1)} = \boldsymbol{H}_1(\boldsymbol{G}_1^T \boldsymbol{G}_2 \circ \boldsymbol{K}_{11}^E)\boldsymbol{H}_2^T + \boldsymbol{H}_1(\boldsymbol{G}_1^T \boldsymbol{H}_2 \circ \boldsymbol{K}_{11}^E)\boldsymbol{G}_2^T$$
$$+ \boldsymbol{G}_1(\boldsymbol{H}_1^T \boldsymbol{G}_2 \circ \boldsymbol{K}_{11}^E)\boldsymbol{H}_2^T + \boldsymbol{G}_1(\boldsymbol{H}_1^T \boldsymbol{H}_2 \circ \boldsymbol{K}_{11}^E)\boldsymbol{G}_2^T, \tag{25}$$

$$\boldsymbol{\Psi}^{(2)} * \boldsymbol{\Psi}^{(2)} = \boldsymbol{H}_1(\boldsymbol{G}_1^T \boldsymbol{G}_2 \circ \boldsymbol{K}_{22}^E)\boldsymbol{H}_2^T + \boldsymbol{H}_1(\boldsymbol{G}_1^T \boldsymbol{H}_2 \circ \boldsymbol{K}_{22}^E)\boldsymbol{G}_2^T$$
$$+ \boldsymbol{G}_1(\boldsymbol{H}_1^T \boldsymbol{G}_2 \circ \boldsymbol{K}_{22}^E)\boldsymbol{H}_2^T + \boldsymbol{G}_1(\boldsymbol{H}_1^T \boldsymbol{H}_2 \circ \boldsymbol{K}_{22}^E)\boldsymbol{G}_2^T, \tag{26}$$

and $(\boldsymbol{\Psi}^{(1)} \odot \boldsymbol{X}) * \boldsymbol{\Psi}^{(2)} = \boldsymbol{H}_1(\boldsymbol{G}_1^T \boldsymbol{X} \boldsymbol{G}_2 \circ \boldsymbol{K}_{12}^E)\boldsymbol{H}_2^T + \boldsymbol{H}_1(\boldsymbol{G}_1^T \boldsymbol{X} \boldsymbol{H}_2 \circ \boldsymbol{K}_{12}^E)\boldsymbol{G}_2^T$

$$+ \boldsymbol{G}_1(\boldsymbol{H}_1^T \boldsymbol{X} \boldsymbol{G}_2 \circ \boldsymbol{K}_{12}^E)\boldsymbol{H}_2^T + \boldsymbol{G}_1(\boldsymbol{H}_1^T \boldsymbol{X} \boldsymbol{H}_2 \circ \boldsymbol{K}_{12}^E)\boldsymbol{G}_2^T, \tag{27}$$

where $\circ$ denotes the Hadamard product between matrices.

*Proof.*
**(1).**

We first prove the equality (27). It suffices to show that the $(i, a)$ term in the left part equals to the $(i, a)$ term in the right part.

Let $e_{ik_1}^1, e_{ik_2}^1, ..., e_{ik_{D^i}}^1$ be the edges incident to the node $v_i$ in graph $\mathcal{G}_1$, where $D^i$ is the degree of the node $v_i$. Without loss of generality, we assume the assigned directions of these edges are, for some $1 \leq s \leq D^i$,

$$v_i \to v_{k_1}, v_i \to v_{k_2}, ...., v_i \to v_{k_s} \quad \text{and} \quad v_{k_{s+1}} \to v_i, v_{k_{s+2}} \to v_i, ..., v_{k_{D^i}} \to v_i. \tag{28}$$

Considering the $i$th row, $\boldsymbol{G}_1(v_i, :)$ and $\boldsymbol{H}_1(v_i, :)$, of the matrix $\boldsymbol{G}_1$ and $\boldsymbol{H}_1$, we have

$$\boldsymbol{G}_1(v_i, e^1_{ik_1}) = \boldsymbol{G}_1(v_i, e^1_{ik_2}) = ... = \boldsymbol{G}_1(v_i, e^1_{ik_s}) = 1, \tag{29}$$

$$\text{and} \quad \boldsymbol{H}_1(v_i, e^1_{ik_{s+1}}) = \boldsymbol{H}_1(v_i, e^1_{ik_{s+2}}) = ... = \boldsymbol{H}_1(v_i, e^1_{ik_{D^i}}) = 1. \tag{30}$$

Any other item in the $i$th row of the matrix $\boldsymbol{G}_1$ and $\boldsymbol{H}_1$ is zero.

Let $e^2_{C_1 a}, e^2_{C_2 a}, ..., e^2_{C_{D^a} a}$ be the edges incident to the node $v_a$ in graph $\mathcal{G}_2$, where $D^a$ is the degree of the node $v_a$. Without loss of generality, we assume the assigned directions of these edges are, for some $1 \le t \le D^a$,

$$v_{C_1} \to v_a, v_{C_2} \to v_a, ..., v_{C_t} \to v_a \quad \text{and} \quad v_a \to v_{C_{t+1}}, v_a \to v_{C_{t+2}}, ..., v_a \to v_{C_{D^a}}. \tag{31}$$

Considering the $a$th row, $\boldsymbol{G}_2(v_a, :)$ and $\boldsymbol{H}_2(v_a, :)$, of the matrix $\boldsymbol{G}_2$ and $\boldsymbol{H}_2$, we have

$$\boldsymbol{G}_2(v_a, e^2_{C_{t+1} a}) = \boldsymbol{G}_2(v_a, e^2_{C_{t+2} a}) = ... = \boldsymbol{G}_2(v_a, e^2_{C_{D^a} a}) = 1, \tag{32}$$

$$\text{and} \quad \boldsymbol{H}_2(v_a, e^2_{C_1 a}) = \boldsymbol{H}_2(v_a, e^2_{C_2 a}) = ... = \boldsymbol{H}_2(v_a, e^2_{C_t a}) = 1. \tag{33}$$

Any other item in the $a$th row of the matrix $\boldsymbol{G}_2$ and $\boldsymbol{H}_2$ is zero.

We first consider the first term $\boldsymbol{H}_1(\boldsymbol{G}_1^T \boldsymbol{X} \boldsymbol{G}_2 \circ \boldsymbol{K}_{12}^E)\boldsymbol{H}_2^T$ in $(\boldsymbol{\Psi}^{(1)} \odot \boldsymbol{X}) * \boldsymbol{\Psi}^{(2)}$ (see (27)),

$$[\boldsymbol{H}_1(\boldsymbol{G}_1^T \boldsymbol{X} \boldsymbol{G}_2 \circ \boldsymbol{K}_{12}^E)\boldsymbol{H}_2^T]_{ia} \tag{34}$$

$$=[\boldsymbol{H}_1(v_i, :)](\boldsymbol{G}_1^T \boldsymbol{X} \boldsymbol{G}_2 \circ \boldsymbol{K}_{12}^E)[\boldsymbol{H}_2(v_a, :)]^T \tag{35}$$

$$=\sum_{\alpha=1}^{D^i} \sum_{\beta=1}^{D^a} \boldsymbol{H}_1(v_i, e^1_{ik_\alpha}) \boldsymbol{H}_2(v_a, e^2_{C_\beta a})[\boldsymbol{G}_1^T \boldsymbol{X} \boldsymbol{G}_2 \circ \boldsymbol{K}_{12}^E](e^1_{ik_\alpha}, e^2_{C_\beta a}) \tag{36}$$

$$=\sum_{\alpha=s+1}^{D^i} \sum_{\beta=1}^{t} [\boldsymbol{G}_1^T \boldsymbol{X} \boldsymbol{G}_2 \circ \boldsymbol{K}_{12}^E](e^1_{ik_\alpha}, e^2_{C_\beta a}) \quad \text{By the definition of } \boldsymbol{H}_1(30), \boldsymbol{H}_2(33) \tag{37}$$

$$=\sum_{\alpha=s+1}^{D^i} \sum_{\beta=1}^{t} [\boldsymbol{G}_1^T \boldsymbol{X} \boldsymbol{G}_2](e^1_{ik_\alpha}, e^2_{C_\beta a}) \times [\boldsymbol{K}_{12}^E](e^1_{ik_\alpha}, e^2_{C_\beta a}) \tag{38}$$

$$=\sum_{\alpha=s+1}^{D^i} \sum_{\beta=1}^{t} [\boldsymbol{K}_{12}^E](e^1_{ik_\alpha}, e^2_{C_\beta a})\big[\boldsymbol{G}_1(:, e^1_{ik_\alpha})^T \boldsymbol{X} \boldsymbol{G}_2(:, e^2_{C_\beta a})\big] \tag{39}$$

$$=\sum_{\alpha=s+1}^{D^i} \sum_{\beta=1}^{t} \boldsymbol{X}_{k_\alpha C_\beta} [\boldsymbol{K}_{12}^E](e^1_{ik_\alpha}, e^2_{C_\beta a}), \tag{40}$$

$$=\sum_{\alpha=1}^{s} \sum_{\beta=t+1}^{D^a} \boldsymbol{X}_{k_\alpha C_\beta} k^E(\vec{q}^1_{ik_\alpha}, \vec{q}^2_{C_\beta a}). \tag{41}$$

where $\boldsymbol{G}_1(:, e^1_{ik_\alpha})$ is a column of $\boldsymbol{G}_1$, corresponding to the edge $e^1_{ik_\alpha}$, and $\boldsymbol{G}_2(:, e^2_{C_\beta a})$ is a column of $\boldsymbol{G}_2$, corresponding to the edge $e^2_{C_\beta a}$. By the definition of $\boldsymbol{G}_1$ (29) and $\boldsymbol{G}_2$ (32), $\boldsymbol{G}_1(v_{k_\alpha}, e^1_{ik_\alpha}) = 1$, since the direction of edge $e^1_{ik_\alpha}$ is assigned as $v_{k_\alpha} \to v_i$, for $\alpha = s+1, s+2, .., D^i$. We also have $\boldsymbol{G}_2(v_{C_\beta}, e^2_{C_\beta a}) = 1$, since the direction of edge $e^2_{C_\beta a}$ is assigned as $v_{C_\beta} \to v_a$, for $\beta = 1, 2, ..., t$. All the other terms in $\boldsymbol{G}_1(:, e^1_{ik_\alpha})$ and $\boldsymbol{G}_2(:, e^2_{C_\beta a})$ are zero. The above discussion explains why the equality (40) holds.

Similarly, we can prove that

$$[\boldsymbol{H}_1(\boldsymbol{G}_1^T \boldsymbol{X} \boldsymbol{H}_2 \circ \boldsymbol{K}_{12}^E)\boldsymbol{G}_2^T]_{ia} = \sum_{\alpha=s+1}^{D^i} \sum_{\beta=t+1}^{D^a} \boldsymbol{X}_{k_\alpha C_\beta} k^E(\vec{q}_{ik_\alpha}^1, \vec{q}_{C_\beta a}^2), \tag{42}$$

$$[\boldsymbol{G}_1(\boldsymbol{H}_1^T \boldsymbol{X} \boldsymbol{G}_2 \circ \boldsymbol{K}_{12}^E)\boldsymbol{H}_2^T]_{ia} = \sum_{\alpha=1}^{s} \sum_{\beta=1}^{t} \boldsymbol{X}_{k_\alpha C_\beta} k^E(\vec{q}_{ik_\alpha}^1, \vec{q}_{C_\beta a}^2), \tag{43}$$

$$[\boldsymbol{G}_1(\boldsymbol{H}_1^T \boldsymbol{X} \boldsymbol{H}_2 \circ \boldsymbol{K}_{12}^E)\boldsymbol{G}_2^T]_{ia} = \sum_{\alpha=1}^{s} \sum_{\beta=t+1}^{D^a} \boldsymbol{X}_{k_\alpha C_\beta} k^E(\vec{q}_{ik_\alpha}^1, \vec{q}_{C_\beta a}^2). \tag{44}$$

Adding (34), (42), (43), and (44), we can obtain

$$\begin{aligned}
&\big[\boldsymbol{H}_1(\boldsymbol{G}_1^T \boldsymbol{X} \boldsymbol{G}_2 \circ \boldsymbol{K}_{12}^E)\boldsymbol{H}_2^T + \boldsymbol{H}_1(\boldsymbol{G}_1^T \boldsymbol{X} \boldsymbol{H}_2 \circ \boldsymbol{K}_{12}^E)\boldsymbol{G}_2^T + \boldsymbol{G}_1(\boldsymbol{H}_1^T \boldsymbol{X} \boldsymbol{G}_2 \circ \boldsymbol{K}_{12}^E)\boldsymbol{H}_2^T + \\
&\boldsymbol{G}_1(\boldsymbol{H}_1^T \boldsymbol{X} \boldsymbol{H}_2 \circ \boldsymbol{K}_{12}^E)\boldsymbol{G}_2^T\big]_{ia} = \sum_{e_{ik}^1 \in \mathcal{E}_1, e_{ca}^2 \in \mathcal{E}_2} \boldsymbol{X}_{kc} k^E(\vec{q}_{ik}^1, \vec{q}_{ca}^2) = [(\boldsymbol{\Psi}^{(1)} \odot \boldsymbol{X}) * \boldsymbol{\Psi}^{(2)}]_{ia},
\end{aligned} \tag{45}$$

where the equality holds because of the fact that $e_{ik_\alpha}^1 \in \mathcal{E}_1$, $\alpha = 1, 2, ..., D^i$, and $e_{C_\beta a}^2 \in \mathcal{E}_2$, $\beta = 1, 2, ..., D^a$.

We finish the proof of the equality (27)!

**(2).**

For proving the equality (25) and (26), we can write

$$\boldsymbol{\Psi}^{(1)} * \boldsymbol{\Psi}^{(1)} = (\boldsymbol{\Psi}^{(1)} \odot \boldsymbol{I}) * \boldsymbol{\Psi}^{(1)} \quad \text{and} \quad \boldsymbol{\Psi}^{(2)} * \boldsymbol{\Psi}^{(2)} = (\boldsymbol{\Psi}^{(2)} \odot \boldsymbol{I}) * \boldsymbol{\Psi}^{(2)}, \tag{46}$$

and directly apply the proved equality (27). $\qquad \square$

## 2 Proofs of the theoretical results of the EnFW algorithm

The EnFW algorithm is used to solve the following optimization problem

$$\min_{\boldsymbol{X}} \quad F_\alpha(\boldsymbol{X}) = J_\alpha(\boldsymbol{X}) + \lambda H(\boldsymbol{X}) \quad \boldsymbol{X} \in \mathcal{D}_n, \tag{47}$$

where $J_\alpha(\boldsymbol{X})$ is a quadratic function with respect to $\boldsymbol{X}$, $H(\boldsymbol{X}) = \sum_{i=1}^{n} \sum_{j=1}^{n} \boldsymbol{X}_{ij} \log(\boldsymbol{X}_{ij})$, $\lambda > 0$, and $\mathcal{D}_n = \{\boldsymbol{X} \in \mathbb{R}_+^{n \times n} | \boldsymbol{X}\vec{\boldsymbol{1}} = \frac{1}{n}\vec{\boldsymbol{1}}, \boldsymbol{X}^T\vec{\boldsymbol{1}} = \frac{1}{n}\vec{\boldsymbol{1}}\}$. Note that $H(\boldsymbol{X})$ is convex with respect to $\boldsymbol{X}$.

Write the quadratic function $J_\alpha(\boldsymbol{X} + s(\boldsymbol{Y} - \boldsymbol{X}))$ as

$$J_\alpha(\boldsymbol{X} + s(\boldsymbol{Y} - \boldsymbol{X})) = J_\alpha(\boldsymbol{X}) + s\langle \nabla J_\alpha(\boldsymbol{X}), \boldsymbol{Y} - \boldsymbol{X} \rangle_{\mathrm{F}} + \frac{1}{2}\mathrm{vec}(\boldsymbol{Y} - \boldsymbol{X})^T \nabla^2 J_\alpha(\boldsymbol{X}) \mathrm{vec}(\boldsymbol{Y} - \boldsymbol{X}) s^2,$$

Define the coefficient of the quadratic term with respect to $s$ as

$$Q(\boldsymbol{X}, \boldsymbol{Y}) \triangleq \frac{1}{2}\mathrm{vec}(\boldsymbol{Y} - \boldsymbol{X})^T \nabla^2 J_\alpha(\boldsymbol{X}) \mathrm{vec}(\boldsymbol{Y} - \boldsymbol{X}) = \frac{1}{2}\langle \nabla J_\alpha(\boldsymbol{Y} - \boldsymbol{X}), \boldsymbol{Y} - \boldsymbol{X} \rangle_{\mathrm{F}}. \tag{48}$$

Define the gap function $g(\boldsymbol{X})$ as

$$g(\boldsymbol{X}) = \langle \nabla J_\alpha(\boldsymbol{X}), \boldsymbol{X} \rangle_{\mathrm{F}} + \lambda H(\boldsymbol{X}) - \min_{\boldsymbol{Y} \in \mathcal{D}_n} \langle \nabla J_\alpha(\boldsymbol{X}), \boldsymbol{Y} \rangle_{\mathrm{F}} + \lambda H(\boldsymbol{Y}). \tag{49}$$

The algorithm is detailed as

### 2.1 Proving proposition 2

**Proposition 2.** *If $\boldsymbol{X}^*$ is an optimal solution of* (47), *then* $g(\boldsymbol{X}^*) = 0$.

**Algorithm 1** The EnFW optimization algorithm for minimizing $F_\alpha$ (47)

1: Initialize $\boldsymbol{X}_0 \in \mathcal{D}_n$
2: **while** not converge **do**
3:    Compute the gradient $\nabla J_\alpha(\boldsymbol{X}_t)$(14),
4:    Obtain the optimal direction $\boldsymbol{Y}_t$, i.e., $\boldsymbol{Y}_t = \operatorname{argmin}_{\boldsymbol{Y}\in\mathcal{D}_n} \langle \nabla J_\alpha(\boldsymbol{X}_t), \boldsymbol{Y}\rangle_{\mathrm{F}} + \lambda H(\boldsymbol{Y})$,
5:    Compute $G_t = g(\boldsymbol{X}_t)$ and $Q_t = Q(\boldsymbol{X}_t, \boldsymbol{Y}_t)$,
6:    Determine the stepsize $s_t$: If $Q_t \leq 0$; $s_t = 1$, else $s_t = \min\{G_t/(2Q_t), 1\}$,
7:    Update $\boldsymbol{X}_{t+1} = \boldsymbol{X}_t + s_t(\boldsymbol{Y}_t - \boldsymbol{X}_t)$.
8: **end**
9: Output the solution $\boldsymbol{X}_\alpha^*$.

*Proof.* It is sufficient to show that

$$\langle \nabla J_\alpha(\boldsymbol{X}^*), \boldsymbol{X}^*\rangle_{\mathrm{F}} + \lambda H(\boldsymbol{X}^*) = \min_{\boldsymbol{Y}\in\mathcal{D}_n} \langle \nabla J_\alpha(\boldsymbol{X}^*), \boldsymbol{Y}\rangle_{\mathrm{F}} + \lambda H(\boldsymbol{Y}).$$

For any $\boldsymbol{Y} \in \mathcal{D}_n$, we have that $\boldsymbol{X}^* + s(\boldsymbol{Y} - \boldsymbol{X}^*) \in \mathcal{D}_n, \forall s \in [0,1]$, because $\mathcal{D}_n$ is a convex set. Since $\boldsymbol{X}^*$ is an optimal solution of (47), we have

$$J_\alpha(\boldsymbol{X}^*) + \lambda H(\boldsymbol{X}^*) \leq J_\alpha(\boldsymbol{X}^* + s(\boldsymbol{Y} - \boldsymbol{X}^*)) + \lambda H(\boldsymbol{X}^* + s(\boldsymbol{Y} - \boldsymbol{X}^*)) \tag{50}$$
$$\leq J_\alpha(\boldsymbol{X}^* + s(\boldsymbol{Y} - \boldsymbol{X}^*)) + \lambda[(1-s)H(\boldsymbol{X}^*) + sH(\boldsymbol{Y})], \tag{51}$$

where the inequality holds because $H(\boldsymbol{X})$ is a convex function. We reorder the above inequality, and get

$$\frac{J_\alpha(\boldsymbol{X}^* + s(\boldsymbol{Y} - \boldsymbol{X}^*)) - J_\alpha(\boldsymbol{X}^*)}{s} \geq \lambda H(\boldsymbol{X}^*) - \lambda H(\boldsymbol{Y}). \tag{52}$$

Taking the limit $s \to 0$ of (52) yields

$$\langle \nabla J_\alpha(\boldsymbol{X}^*), \boldsymbol{Y} - \boldsymbol{X}^*\rangle_{\mathrm{F}} \geq \lambda H(\boldsymbol{X}^*) - \lambda H(\boldsymbol{Y}). \tag{53}$$

Reordering the above equality yields

$$\forall \boldsymbol{Y} \in \mathcal{D}_n, \ \langle \nabla J_\alpha(\boldsymbol{X}^*), \boldsymbol{X}^*\rangle_{\mathrm{F}} + \lambda H(\boldsymbol{X}^*) \leq \langle \nabla J_\alpha(\boldsymbol{X}^*), \boldsymbol{Y}\rangle_{\mathrm{F}} + \lambda H(\boldsymbol{Y}) \tag{54}$$
$$\iff \langle \nabla J_\alpha(\boldsymbol{X}^*), \boldsymbol{X}^*\rangle_{\mathrm{F}} + \lambda H(\boldsymbol{X}^*) = \min_{\boldsymbol{Y}\in\mathcal{D}_n} \langle \nabla J_\alpha(\boldsymbol{X}^*), \boldsymbol{Y}\rangle_{\mathrm{F}} + \lambda H(\boldsymbol{Y}). \tag{55}$$

$\square$

## 2.2 Proving Theorem 1

**Theorem 1.** *The generated objective function value sequence, $\{F_\alpha(\boldsymbol{X}_t)\}_{t=0}$, will decreasingly converge. The generated points sequence, $\{\boldsymbol{X}_t\}_{t=0} \subseteq \mathcal{D}_n \subseteq \mathbb{R}^{n\times n}$, will weakly converge to the first-order stationary point, at the rate $O(\frac{1}{\sqrt{t+1}})$, i.e,*

$$\min_{1\leq t\leq T} g(\boldsymbol{X}_t) \leq \frac{2\max\{\triangle_0, \sqrt{L\triangle_0/n}\}}{\sqrt{T+1}}, \tag{56}$$

*where $\triangle_0 = F_\alpha(\boldsymbol{X}_0) - \min_{X\in\mathcal{D}_n} F_\alpha(\boldsymbol{X})$, and $L$ is the largest absolute eigenvalue of $\nabla^2 J_\alpha(\boldsymbol{X})$.*

**Before we prove Theorem 1, we introduce a lemma.**

**Lemma 3.** *The generated objective function value sequence, $\{F_\alpha(\boldsymbol{X}_t)\}_{t=0}$, satisfies*

$$F_\alpha(\boldsymbol{X}_{t+1}) \leq F_\alpha(\boldsymbol{X}_t) - (G_t - \frac{L}{n}s)s, \ \forall s \in [0,1]. \tag{57}$$

*Proof.* Since $H(\boldsymbol{X})$ is convex, we have

$$\forall s \in [0,1], \lambda H(\boldsymbol{X}_t + s(\boldsymbol{Y}_t - \boldsymbol{X}_t)) \leq \lambda H(\boldsymbol{X}_t) + s(\lambda H(\boldsymbol{Y}_t) - \lambda H(\boldsymbol{X}_t)). \tag{58}$$

Write the quadratic function $J_\alpha(\boldsymbol{X}_t + s(\boldsymbol{Y}_t - \boldsymbol{X}_t))$ as

$$J_\alpha(\boldsymbol{X}_t + s(\boldsymbol{Y}_t - \boldsymbol{X}_t)) = J_\alpha(\boldsymbol{X}_t) + s\langle \nabla J_\alpha(\boldsymbol{X}_t), \boldsymbol{Y}_t - \boldsymbol{X}_t\rangle_{\mathrm{F}} + Q(\boldsymbol{X}_t, \boldsymbol{Y}_t)s^2. \tag{59}$$

Adding (58) and (59), and reordering the resulting inequality, we have

$$F_\alpha(\boldsymbol{X}_t + s(\boldsymbol{Y}_t - \boldsymbol{X}_t)) \leq F_\alpha(\boldsymbol{X}_t) + s\left[\lambda H(\boldsymbol{Y}_t) - \lambda H(\boldsymbol{X}_t) + \langle \nabla J_\alpha(\boldsymbol{X}_t), \boldsymbol{Y}_t - \boldsymbol{X}_t \rangle_\mathrm{F}\right] + Q(\boldsymbol{X}_t, \boldsymbol{Y}_t)s^2. \tag{60}$$

Since $\lambda H(\boldsymbol{Y}_t) + \langle \nabla J_\alpha(\boldsymbol{X}_t), \boldsymbol{Y}_t \rangle_\mathrm{F} = \min_{\boldsymbol{Y} \in \mathcal{D}_n} \langle \nabla J_\alpha(\boldsymbol{X}_t), \boldsymbol{Y} \rangle_\mathrm{F} + \lambda H(\boldsymbol{Y})$ (See the 4th line in Algorithm 1), we have

$$G_t = g(\boldsymbol{X}_t) = -\left[\lambda H(\boldsymbol{Y}_t) - \lambda H(\boldsymbol{X}_t) + \langle \nabla J_\alpha(\boldsymbol{X}_t), \boldsymbol{Y}_t - \boldsymbol{X}_t \rangle_\mathrm{F}\right], \tag{61}$$

which is based on the definition of $g(\boldsymbol{X})$ (49) and $G_t$ (See the 5th line in Algorithm 1). Substituting (61) into (60), we have

$$\forall s \in [0,1], F_\alpha(\boldsymbol{X}_t + s(\boldsymbol{Y}_t - \boldsymbol{X}_t)) \leq F_\alpha(\boldsymbol{X}_t) - (G_t - Q_t s)s. \tag{62}$$

Set $s = s_t$ (See the 6th and 7th line in Algorithm 1), we have

$$F_\alpha(\boldsymbol{X}_{t+1}) \leq F_\alpha(\boldsymbol{X}_t) - (G_t - Q_t s_t)s_t.$$

Now we consider the function

$$A_t(s) = (G_t - Q_t s)s, s \in [0,1].$$

We discuss the maximizer of $A_t(s)$ for $s \in [0,1]$. Considering that $G_t$ is nonnegative (this is because of the definition of $g(\boldsymbol{X})$), we have

1. If $Q_t \leq 0$, then $A_t(s)$ achieve its maximum at $s = 1$,

2. If $Q_t > 0$, then $A_t(s)$ achieve its maximum at $s = \min\{G_t/(2Q_t), 1\}$.

Therefore, our stepsize $s_t$ is just the maximizer of $A_t(s)$. That is

$$\begin{aligned} F_\alpha(\boldsymbol{X}_{t+1}) \leq F_\alpha(\boldsymbol{X}_{t+1}) \leq &F_\alpha(\boldsymbol{X}_t) - (G_t - Q_t s_t)s_t = F_\alpha(\boldsymbol{X}_t) - \max_{s \in [0,1]}(G_t - Q_t s)s \\ \leq &F_\alpha(\boldsymbol{X}_t) - (G_t - Q_t s)s \quad \forall s \in [0,1]. \end{aligned} \tag{63}$$

Since

$$Q_t = Q(\boldsymbol{X}_t, \boldsymbol{Y}_t) = \frac{1}{2}\mathrm{vec}(\boldsymbol{Y}_t - \boldsymbol{X}_t)^T \nabla^2 J_\alpha(\boldsymbol{X}_t)\mathrm{vec}(\boldsymbol{Y}_t - \boldsymbol{X}_t) \leq \frac{L}{2}\|\boldsymbol{X}_t - \boldsymbol{Y}_t\|_\mathrm{F}^2,$$

and

$$\|\boldsymbol{X} - \boldsymbol{Y}\|_\mathrm{F}^2 \leq \frac{2}{n}, \forall \boldsymbol{X}, \boldsymbol{Y} \in \mathcal{D}_n,$$

we have $Q_t \leq \frac{L}{n}$.

Combining (63) and the fact $Q_t \leq \frac{L}{n}$, we obtain the desired result. $\qquad \square$

**Now we prove Theorem 1**

*Proof.* We consider the inequality (57) in Lemma 3.

If $G_t > \frac{2L}{n}$, then $(G_t - \frac{L}{n}s)s$ is maximized at $s = 1$. So

$$F_\alpha(\boldsymbol{X}_{t+1}) \leq F_\alpha(\boldsymbol{X}_t) - (G_t - \frac{L}{n}) \leq F_\alpha(\boldsymbol{X}_t) - \frac{G_t}{2},$$

where the last equality holds because $\frac{L}{n} \leq \frac{G_t}{2}$.

If $G_t \leq \frac{2L}{n}$, then $(G_t - \frac{L}{n}s)s$ is maximized at $s = \frac{nG_t}{2L}$. So

$$F_\alpha(\boldsymbol{X}_{t+1}) \leq F_\alpha(\boldsymbol{X}_t) - (\frac{G_t}{2})\frac{nG_t}{2L}.$$

In summary,

$$F_\alpha(\boldsymbol{X}_{t+1}) \leq F_\alpha(\boldsymbol{X}_t) - \frac{G_t}{2}\min\{1, \frac{nG_t}{2L}\} \tag{64}$$

**(I.)** Since $G_t \geq 0$ by definition (49), we have $F_\alpha(\boldsymbol{X}_{t+1}) \leq F_\alpha(\boldsymbol{X}_t)$, i.e., $\{F_\alpha(\boldsymbol{X}_t)\}_{t=0}$ is a decreasing sequence. $F_\alpha(\boldsymbol{X})$ is continuous on the compact region $\mathcal{D}_n$, which implies that $F_\alpha(\boldsymbol{X})$ is bounded below. So the sequence $\{F_\alpha(\boldsymbol{X}_t)\}_{t=0}$ will converge [3].

**(II.)** Taking the sum of (64) over $t = 0, 1, 2, ..., T$, we obtain,

$$F_\alpha(\boldsymbol{X}_{T+1}) - F_\alpha(\boldsymbol{X}_0) \leq -\sum_{t=0}^{T} \frac{G_t}{2} \min\{1, \frac{G_t n}{2L}\}. \tag{65}$$

Let

$$G_T^* = \min_{0 \leq t \leq T} G_t = \min_{0 \leq t \leq T} G(\boldsymbol{X}_t).$$

Considering the additional fact that $-\triangle_0 \triangleq F_\alpha(\boldsymbol{X}^*) - F_\alpha(\boldsymbol{X}_0) \leq F_\alpha(\boldsymbol{X}_{T+1}) - F_\alpha(\boldsymbol{X}_0)$, we have

$$\triangle_0 \geq \sum_{t=0}^{T} \frac{G_t}{2} \min\{1, \frac{G_t n}{2L}\} \geq (T+1)\frac{G_T^*}{2} \min\{1, \frac{G_T^* n}{2L}\}.$$

**(a.)** If $\frac{nG_T^*}{2L} \geq 1$, then $\triangle_0 \geq (T+1)\frac{G_T^*}{2} \iff G_T^* \leq \frac{2\triangle_0}{T+1} \leq \frac{2\triangle_0}{\sqrt{T+1}}$;

**(b.)** If $\frac{nG_T^*}{2L} < 1$, then $\triangle_0 \geq (T+1)(\frac{G_T^*}{2})\frac{nG_T^*}{2L} \iff G_T^* \leq \sqrt{\frac{4L\triangle_0}{n(T+1)}}$.

In summary, we have $G_T^* \leq \frac{2\max\{\triangle_0, \sqrt{L\triangle_0/n}\}}{\sqrt{T+1}}$. $\qquad\square$

## 2.3 Proving Theorem 2

**Theorem 2.** *If $\boldsymbol{J}_\alpha(\boldsymbol{X})$ is convex, we have $F_\alpha(\boldsymbol{X}_T) - F_\alpha(\boldsymbol{X}^*) \leq \frac{4L}{n(T+1)}$.*

**Before we prove Theorem 2, we introduce a lemma.**

**Lemma 4.** *If $\boldsymbol{J}_\alpha(\boldsymbol{X})$ is convex, and let $\boldsymbol{X}^*$ is a global minimizer of problem (47), then $g(\boldsymbol{X}_t) \geq F_\alpha(\boldsymbol{X}_t) - F_\alpha(\boldsymbol{X}^*)$.*

*Proof.*

$$g(\boldsymbol{X}_t) = \langle \nabla J_\alpha(\boldsymbol{X}_t), \boldsymbol{X}_t \rangle_{\mathrm{F}} + \lambda H(\boldsymbol{X}_t) - \min_{\boldsymbol{Y} \in \mathcal{D}_n} \langle \nabla J_\alpha(\boldsymbol{X}_t), \boldsymbol{Y} \rangle_{\mathrm{F}} + \lambda H(\boldsymbol{Y}). \tag{66}$$

$$= \max_{\boldsymbol{Y} \in \mathcal{D}_n} \left\{ \langle \nabla J_\alpha(\boldsymbol{X}_t), \boldsymbol{X}_t - \boldsymbol{Y} \rangle_{\mathrm{F}} + \lambda H(\boldsymbol{X}_t) - \lambda H(\boldsymbol{Y}) \right\} \tag{67}$$

$$\geq \langle \nabla J_\alpha(\boldsymbol{X}_t), \boldsymbol{X}_t - \boldsymbol{X}^* \rangle_{\mathrm{F}} + \lambda H(\boldsymbol{X}_t) - \lambda H(\boldsymbol{X}^*) \tag{68}$$

$$\geq J_\alpha(\boldsymbol{X}_t) - J_\alpha(\boldsymbol{X}^*) + \lambda H(\boldsymbol{X}_t) - \lambda H(\boldsymbol{X}^*) \tag{69}$$

$$= F_\alpha(\boldsymbol{X}_t) - F_\alpha(\boldsymbol{X}^*), \tag{70}$$

where the inequality (69) holds because $J_\alpha(\boldsymbol{X})$ is convex. $\qquad\square$

**Now we prove Theorem 2.**

*Proof.* We still need the inequality (57) in Lemma 3.

$$F_\alpha(\boldsymbol{X}_{t+1}) \leq F_\alpha(\boldsymbol{X}_t) - (G_t - \frac{L}{n}s)s$$

$$\leq F_\alpha(\boldsymbol{X}_t) - [F_\alpha(\boldsymbol{X}_t) - F_\alpha(\boldsymbol{X}^*)]s + \frac{L}{n}s^2, \ \forall s \in [0, 1], \tag{71}$$

where the inequality (71) holds because of Lemma 4. Reordering (71) yields

$$[F_\alpha(\boldsymbol{X}_{t+1}) - F_\alpha(\boldsymbol{X}^*)] \leq (1-s)[F_\alpha(\boldsymbol{X}_t) - F_\alpha(\boldsymbol{X}^*)] + \frac{L}{n}s^2. \tag{72}$$

We set $s = \frac{2}{t+1}$, and multiply $\frac{t(t+1)}{2}$ on both sides. Then we have

$$\frac{t(t+1)}{2}\big[F_\alpha(\boldsymbol{X}_{t+1}) - F_\alpha(\boldsymbol{X}^*)\big] \leq \frac{t(t-1)}{2}\big[F_\alpha(\boldsymbol{X}_t) - F_\alpha(\boldsymbol{X}^*)\big] + \frac{L}{n}\frac{2t}{t+1}$$

$$\leq \frac{t(t-1)}{2}\big[F_\alpha(\boldsymbol{X}_t) - F_\alpha(\boldsymbol{X}^*)\big] + \frac{2L}{n} \tag{73}$$

Taking the sum of (73) over $t = 1, 2, ..., T$, we have

$$\frac{T(T+1)}{2}\big[F_\alpha(\boldsymbol{X}_{T+1}) - F_\alpha(\boldsymbol{X}^*)\big] \leq \frac{2TL}{n} \iff F_\alpha(\boldsymbol{X}_{T+1}) - F_\alpha(\boldsymbol{X}^*) \leq \frac{4L}{n(T+1)}. \tag{74}$$

We obtain the desired result. $\qquad\square$

# 3 Experiments

In the first part, we present the implementation details of the experiments in the paper, including the description of Sinkhorn–Knopp algorithm, and convergence criterions for EnFW.

## 3.1 Implementation details

**The Sinkhorn-Knopp algorithm** [4, 1] is used to solve the problem

$$\min_{\boldsymbol{X}} \langle \boldsymbol{D}, \boldsymbol{X} \rangle_{\mathrm{F}} + \lambda H(\boldsymbol{X}) \quad \text{s.t.} \quad \boldsymbol{X} \geq 0, \quad \boldsymbol{X}\vec{\mathbf{1}}_n = \vec{\boldsymbol{a}}, \text{ and } \boldsymbol{X}^T\vec{\mathbf{1}}_n = \vec{\boldsymbol{b}}, \tag{75}$$

where $\vec{\boldsymbol{a}}$ and $\vec{\boldsymbol{b}}$ satisfy $\sum_{i=1}^n \vec{\boldsymbol{a}}_i = \sum_{j=1}^n \vec{\boldsymbol{b}}_j = 1$. In our case $\vec{\boldsymbol{a}} = \frac{1}{n}\vec{\mathbf{1}}_n$ and $\vec{\boldsymbol{b}} = \frac{1}{n}\vec{\mathbf{1}}_n$.

The algorithm is shown in Algorithm 2. The detailed derivations can be found in [1]. In our

---

**Algorithm 2** The Sinkhorn-Knopp optimization algorithm for minimizing (75)

---

1: Initialize $\vec{\boldsymbol{r}} = \frac{1}{n}\vec{\mathbf{1}}_n$, $t = 0$, and RelativeError $= 1$, and write $\boldsymbol{C} = \exp(-\frac{\boldsymbol{D}}{\lambda})$ (pointwise).
2: **while** $t <$ MaxNum$_{\text{Sinkhorn}}$ and RelativeError $>$ Tolerance **do**
3: $\quad \vec{\boldsymbol{r}}_{\text{new}} = (\frac{1}{n}\vec{\mathbf{1}}_n)./\Big\{\boldsymbol{C}\big[(\frac{1}{n}\vec{\mathbf{1}}_n)./(\boldsymbol{C}^T\vec{\boldsymbol{r}})\big]\Big\}$.
4: $\quad$ RelativeError $= \frac{\|\vec{\boldsymbol{r}}_{\text{new}} - \vec{\boldsymbol{r}}\|_2}{\|\vec{\boldsymbol{r}}\|_2}$.
5: $\quad t = t + 1$
6: **end**
7: $\vec{\boldsymbol{s}} = (\frac{1}{n}\vec{\mathbf{1}}_n)./(\boldsymbol{C}^T\vec{\boldsymbol{r}})$, $\boldsymbol{X}^* = \text{diag}(\vec{\boldsymbol{r}})\boldsymbol{C}\text{diag}(\vec{\boldsymbol{r}})$.
8: Output the solution $\boldsymbol{X}^*$.

---

experiments, we set MaxNum$_{\text{Sinkhorn}} = 10000$, and set Tolerance $= 10^{-6}$.

**Convergence**
fied for continuing the iterations in Algorithm 1.

1. The maximal number of iterations is set to be 1000, i.e., $t < 1000$;

2. The tolerance of relative error between $\boldsymbol{X}_{t+1}$ and $\boldsymbol{X}_t$ is set to be $10^{-8}$, i.e, $\frac{\|\boldsymbol{X}_{t+1} - \boldsymbol{X}_t\|_{\mathrm{F}}}{\|\boldsymbol{X}_t\|_{\mathrm{F}}} > 10^{-8}$.

3. The gap $G_t = g(\boldsymbol{X}_t)$ should be greater than $eps$ to avoid numerical error, where $eps$ is the spacing of floating point numbers of Matlab.

## 3.2 Experimental results

We re-implement the experiments in Section 6.1 in the paper, for comparing the matching performance of different algorithms. Here, we test the performance of KerGM$_{\mathrm{I}}$ on synthetic datasets. That is, we use the exact edge affinity kernel[1] and compute the gradients by using the formulas in (25), (26), and (27). We still consider the same parameter settings:

Figure 2: Comparison of graph matching algorithms on synthetic datasets

1. We change the number of outlier nodes, $n_{\text{out}}$, from 0 to 50 while fixing the noise, $\sigma = 0$, and the edge density, $\rho = 1$.

2. We change $\sigma$ from 0 to 0.2 while fixing $n_{\text{out}} = 0$ and $\rho = 1$.

3. We change $\rho$ from 0.3 to 1 while fixing $n_{\text{out}} = 5$ and $\sigma = 0.1$.

We repeat the experiments 100 times, and report the average matching accuracies and objective function values. In Fig. 2, we show the results. It can be seen that our $\text{KerGM}_{\text{I}}$ still consistently outperforms the baselines.

## Footnotes

[1]Note that in the paper, we test the performance of KerGM$_{\mathrm{II}}$ on synthetic datasets. We in fact use the Fourier random features [2] to approximate the kernel.