[Reviews · NeurIPS 2019]

Reviewer 1



We found that the paper presents interesting results and it is well written. We are not expert in this topic. However, we found a couple of issues in the paper. - In the text, it is written “Koopman”. It should be Koopmanns. - The authors often emphasize on a particular finding in the paper, which is that the connection between Lawler’s QAP and Koopman-Beckmann’s QAP. It turns out that the connections between the two formulations are straightforward and well known in the literature. See for example: RE Burkard, E Cela, PM Pardalos, LS Pitsoulis, The quadratic assignment problem. Handbook of combinatorial optimization, pp 1713-1809, 1998 - in [42], the authors proposed a generalization of the graph matching, where the inner-product of data is replaced by a general function. This is pretty related to the use of kernels. While [42] lacks the theoretical foundations that allows the RKHS in the submitted paper, the authors of the latter should well explain the differences with [42]. - Kernels to measure similarities between edges, or between nodes, have been previously investigated in the literature. Moreover, the path-following strategy used for convex-concave relaxations is also not new. We think that the authors need to emphasize on the contributions of this work. ================================= We thank the authors for the rebuttal, which responded to all our concerns. We have updated our review correspondingly.

Reviewer 2



- The authors present a kernelized version of the PATH algorithm for network alignment, utilizing a novel EnFW algorithm to approximately solve successive Frank-Wolfe iterations. This allows for dramatic algorithmic speed-up versus the standard Hungarian algorithm approach to F-W. - $K$ in (3) only scales quadruply with respect to $n$ in the dense graph setting. In sparse settings, if the max degree is polylog(n), then up to a polylog factor, $K$ only has $n^2$ nonzero elements and can be efficiently computed and stored. In the dense setting, the gain from kernelizing is more substantial. - The EnFW subroutine is quite an interesting alternative to the Hungarian algorithm; does EnFW have a provable runtime? Introducing the entropy regularizing term allows for dramatic speed-ups; does it affect performance versus an exact solution to (14) being computed at each step? There are a number of variants of F-W that scale to large problems (see, for example, Ding, L., & Udell, M. (2018). Frank-Wolfe Style Algorithms for Large Scale Optimization. In Large-Scale and Distributed Optimization (pp. 215-245). ). It would be interesting to see EnFW compared (runtime-wise) to these existing approximate F-W solvers. - On page 5, you claim: We give an explicit formula for computing the stepsize s, instead of making a linear search on [0, 1] for optimizing $F_{\alpha} (X + s(Y − X ))$. In classic F-W applied to GM, $J_\alpha$ is a quadratic function that can also be exactly optimized. - For the simulation and image experiments, you compare KerGM to 5 algorithms, the most recent of which is from 2012. There have been a large number of GM algorithms in the recent literature that scale to very large graphs (see, for example, Zhang, S., & Tong, H. (2016, August). Final: Fast attributed network alignment. In Proceedings of the 22nd ACM SIGKDD International Conference on Knowledge Discovery and Data Mining (pp. 1345-1354). ACM. and Heimann, M., Shen, H., Safavi, T., & Koutra, D. (2018, October). Regal: Representation learning-based graph alignment. In Proceedings of the 27th ACM International Conference on Information and Knowledge Management (pp. 117-126). ACM. ). Including comparisons to more recent approaches would be helpful for gauging the efficacy of your current algorithm. The experiments on the PPI networks are more compelling for including more recent comparisons. - In the synthetic datasets experiment, what are the outlier nodes? - Line 140 in the appendix: Why is the final condition not $G_T^*\leq \frac{2\Delta_0}{T+1}$? Where does the $\sqrt{\cdot}$ come from? This would have a small impact on the statement of Theorem 1.

Reviewer 3



# Update after rebuttal I thank the authors for their rebuttal, which addressed my questions. I am confident that this would make a good submission; I am thus raising my score to '8'. # Summary of the review This is a very good paper, which I enjoyed reading. The write-up is very dense and technical, but the main message is very clear. I particularly enjoyed the unifying view on the two QAP that is given by the paper. It is a good use of kernel theory. I have a number of questions and suggestions to further improve the paper, though, as well as some comments about the clarity of the proposed method. # Review ## Method (clarity, technical correctness) - In l. 25, the affinity matrix is introduced as having a dimensionality of $n^2 \times n^2$. At this point, it is not clear why that should be the case; ideally, the introduction should lay the ground for giving a good explanation about the computational complexity. - While the paper makes it clear later on, the use of the term 'array' struck me as unclear when first reading it. This should be explained at the beginning, ideally. - Mentioning the 'local maxima issue' should be explained a little bit; it becomes clear later on in the paper but since I am not an expert in QAP (my background being pure mathematics), it took me some time to get that concept. - The node and edge similarity functions in l. 76 should be briefly introduced. Later on, they are taken to be kernels; could this not already be done here? I do not see a need for assumption 1; I think the paper could just introduce the node and edge similarity measurements as kernel functions directly. - The explanation of kernel functions in Section 2.2 could be improved. Why not use the explanation of positive definite matrices here? - In l. 118--120, I am not sure about the terminology: should the equality in the equation not be the inner product in $F_{\mathcal{H}_k}$ instead of $F_{\mathcal{H}}$ - In Eq. 5, what is $K^P$? The notation looks somewhat similar to $\mathcal{P}$, so I would assume that the matrix 'belongs' to that space as well. - A notational suggestion: would it be easier to write $\Psi^{(1)}$ instead of $\Psi^1$? While it should be clear from the context, one might confuse the second notation with an exponential/multiplication operation. - The two relaxations in Eq. 6 and Eq. 7 should be explained better. Why not briefly prove convexity and concavity, respectively? - The explanation of the computational complexity (and the improvements obtained by the new strategy) should be made more clear. - For the scalability comparison in l. 239ff., would it be possible to use a normalised comparison, for example by dividing the times of each method by the respective time of the smallest method? I would like some more details here because it is not clear whether it is justified to compare different codes that might not be optimised for a certain architecture. - The analysis of parameter sensitivity seems to indicate that there is a larger variance for $\lambda$. Ideally, I would like to see more data points in Figure 3 to demonstrate the behaviour with respect to this parameter. - Why are the experiments from Section 6.2 not repeated multiple times? I would be interested in knowing the standard deviations or variances. - How have the parameters in Section 6.3 for the heat kernel signature been selected? Would it matter if the values are changed there? ## Experimental setup The experimental setup seems to be sufficient for me. I already listed my concerns about doing a more thorough parameter analysis. Other than that, I have no further comments---I should stress that I am not super familiar with competitor methods. ## Language & style The paper is well written for the most part, but certain small oversights make it harder to understand at times. I would suggest another detailed pass. - l. 27: 'based a very' --> 'based on a very' - l. 30: 'two arrays in reproducing kernel Hilbert space' --> 'two arrays in a reproducing kernel Hilbert space' - l. 32: 'develop path-following strategy' --> 'develop a path-following strategy' - l. 38: 'years, myriad graph' --> 'years, a myriad of graph' - l. 49: 'the author' --> 'the authors' - l. 86: 'there exist a' --> 'there exists a' - l. 121: 'Laweler's QAP' --> 'Lawler's QAP' - l. 146: 'small value of $D$' --> 'small values of $D$' - l. 158: 'may need to call the Hungarian' --> 'may require calling the Hungarian' A small 'pet peeve' of mine: I would refrain from using citations as nouns. Instead of writing 'In [2, 14, 21]', I would maybe write 'Previous work [2, 14, 21]'. Moreover, I would suggest making sure that citations are always sorted upon citing them: In L. 43, for example, the citation '[29, 28]' should be written as '[28, 29]'. In l. 116, the citations '[45], [19]' should be merged into one citation. One issue I have is the length of the paper, or rather the length of the paper and the supplementary materials. Ideally, the paper would give a small idea about relevant proofs. At present, to fully understand the paper, one has to read all the supplementary materials and the main text. Ideally, the paper could give at least some proof ideas. ## Bibliography To my understanding, all relevant related work has been cited. Nonetheless, in the interest of improving this paper even further, I would suggest improving the consistency: - Make sure that the capitalisation of names of conferences and journals is consistent. For example, 'International journal of pattern recognition...' should be called 'International Journal of Pattern Recognition', and so on. - Make sure that authors names are used consistently. For example, in item 12, the author names look very incorrect.

[Author Response · NeurIPS 2019]

# KerGM: Kernelized Graph Matching (Paper ID:1849)

## To Reviewer 2

**Q1:** Connection between the Koopmanns-Beckman QAP (KB-QAP) and the Lawler QAP(L-QAP).

**A1:** In the literature, it's well-known that KB-QAP can be written in the form of L-QAP, which happens only when edge attributes are scalar and their similarity function is simple multiplication. In our work, we consider the inverse direction. That is, by introducing $\mathcal{H}-$operations, we can write the L-QAP in the KB's form, i.e., the KB alignment between Hilbert arrays, allowing graphs with complex (e.g., vectorial) edge attributes. The KB's alignment form gives us effective convex-concave relaxations, and significant reduction of the time and space complexity of the L-QAP.

**Q2:** Difference with the work [42].

**A2:** Previous work [42] replaces the discrete term $X_{ia}X_{jb}$ with a family of continuous functions $f_\delta$ to gradually solve the discrete L-QAP, where $f_\delta$ relates to kernels. However, in our settings, the edge affinity terms $K_{ij,ab}$ (notation $A_{ij:ab}$ in [42]) are kernel values, which leads to significantly different strategies of solving L-QAP, e.g., how to make relaxations. One important advantage of our work is that our algorithm can scale to dense graphs with thousands of nodes ($n = O(10^3)$), while work [42] cannot because it requires computing the $n^2 \times n^2$ matrix $K$ (notation $A$ in [42]).

**Q3:** Contributions of this work.

**A3:** As summarized by Reviewer 3 and 4, we developed **(1).** a unified perspective of two QAPs, **(2).** an efficient entropy-regularized Frank-Wolfe algorithm for optimization, **(3).** the KerGM framework for solving graph matching problems. We agree that kernels are popular in measuring similarities. However, we believe this is the first work of solving the large-scale Lawler's QAP by aligning arrays in RKHS. There exist works that leverage the path-following strategy to solve L-QAP, and they differ with each other in how to relax the original problem. We proposed totally new convex-concave relaxations, and more notably, we made it scalable to large graphs.

Figure A: (a) an example of (in)outliers; (b) comparing accuracies; (c) sensitivity of $\lambda$; (d) comparing accuracies on Pascal dataset.

## To Reviewer 3

**Q1: (1).** Provable runtime of the EnFW subroutine. **(2).** How does the regularizer affect performance?

**A1: (1).** At each step, the Sinkhorn-Knopp algorithm converges at the linear rate, i.e., $0 < \limsup \|X_{k+1} - X^*\|/\|X_k - X^*\| < 1$, as proved in the Section 4 of the paper "George W.Soules, The rate of convergence of Sinkhorn balancing". The outer iteration convergence rates are shown in Theorem 1 and 2 in our paper. **(2).** We re-conduct the second random graph matching experiments with fixed outliers and edge density, and varying noise. In Fig. A(b), we compare EnFW ($\lambda = 0.005$) and the exact FW ($\lambda = 0$) that uses the Hungarian algorithm during each outer iteration, both of which are under our KerGM formulation. The exact FW performs slightly better than EnFW.

**Q2:** Compare "KerGM" with "Final" (Zhang et al. 2016) and "Regal" (Heimann et al. 2018)

**A2:** In Fig. A(b), we show the results of "Final". The standard experimental protocol may be not suitable for "Final" since it doesn't directly solve the Lawler's QAP. For "Regal", we are still trying to fit the code in our settings.

**Q3:** Definition of the outliers. **A3:** We show the concept of outliers in Fig. A(a), where $G$ and $G'$ are weighted graphs.

**Q4:** Where does the $\sqrt{\cdot}$ come from in Line 140 in the supplementary material?

**A4:** Thanks for the careful review. We are sorry that there is a little mistake. Line 140 should be changed into $\Delta_0 \geq (T+1)G_T^*/2 \Rightarrow G_T^* \leq 2\Delta_0/(T+1) \leq 2\Delta_0/\sqrt{T+1}$. The statement in Theorem 1 becomes $G_T^* \leq 2\max\{\Delta_0, \sqrt{L\Delta_0/n}\}/\sqrt{T+1}$.

## To Reviewer 4

**A1:** What is $K^P$? **Q1:** We are sorry that it's a typo. It should be $K^N$, the node attributes affinity matrix.

**Q2:** Explanation on computational complexity

**A2:** Existing methods require pre-computing the affinity matrix $K \in \mathbb{R}^{n^2 \times n^2}$, whose both the time and space cost are $O(n^4)$ for dense graphs. In each outer iteration of optimization, the time cost of computing gradient is also $O(n^4)$, because it involves the term $K\mathrm{vec}(X)$. Our KerGM doesn't require pre-computing $K$. With the approximate feature map, the space cost is $O(Dn^2)$ and the time cost of computing gradients is $O(Dn^3)$, where $D << n$ (see Section 4.2).

**Q3 (experiments): (1).** More points for sensitivity of $\lambda$. **(2).** Standard deviations of matching results on Section 6.2.

**A3: (1).** We added more variants of $\lambda$ in Fig. A(c). **(2).** The results in Section 6.2 are the averaged accuracies of graph matching between many pairs of images. We will add the error bars in the revised paper. Fig. A(d) shows an example.

**Q4:** How have the parameters in Section 6.3 been selected? **A4:** We tried different combinations of these parameters, and found that the results are not sensitive to them except for the parameter $\gamma$, where we select from $\{2, 20, 200, 2000\}$.

**A\*:** Thanks for other comments on organization, writing style, and notations. We will follow them in the revised paper.

[Meta-Review · NeurIPS 2019]

This is a very dense but nice paper that provides an original solution to a very important problem (graph matching). The reviewers raised a number of questions, which were all carefully and convincingly addressed in the rebuttal. There is a good balance between theoretical justification of the proposed method, algorithmic contributions, and promising experimental results.